# Enhanced Bilevel Optimization via Bregman Distance

**Feihu Huang[1,2], Junyi Li[1], Shangqian Gao[1], Heng Huang[1]**
[1]Electrical & Computer Engineering, University of Pittsburgh, Pittsburgh, PA, United States
[2]College of Computer Science & Technology, Nanjing University of Aeronautics & Astronautics,
Nanjing, China
huangfeihu2018@gmail.com, junyili.ai@gmail.com, shg84@pitt.edu, heng.huang@pitt.edu

## Abstract

Bilevel optimization has been recently used in many machine learning problems such as hyperparameter optimization, policy optimization, and meta learning. Although many bilevel optimization methods have been proposed, they still suffer from the high computational complexities and do not consider the more general bilevel problems with nonsmooth regularization. In the paper, thus, we propose a class of enhanced bilevel optimization methods with using Bregman distance to solve bilevel optimization problems, where the outer subproblem is nonconvex and possibly nonsmooth, and the inner subproblem is strongly convex. Specifically, we propose a bilevel optimization method based on Bregman distance (BiO-BreD) to solve deterministic bilevel problems, which achieves a lower computational complexity than the best known results. Meanwhile, we also propose a stochastic bilevel optimization method (SBiO-BreD) to solve stochastic bilevel problems based on stochastic approximated gradients and Bregman distance. Moreover, we further propose an accelerated version of SBiO-BreD method (ASBiO-BreD) using the variance-reduced technique, which can achieve a lower computational complexity than the best known computational complexities with respect to condition number $\kappa$ and target accuracy $\epsilon$ for finding an $\epsilon$-stationary point. We conduct data hyper-cleaning task and hyper-representation learning task to demonstrate that our new algorithms outperform related bilevel optimization approaches.

## 1 Introduction

Bilevel optimization can effectively solve the problems with a hierarchical structure, thus it recently has been widely used in many machine learning tasks such as hyper-parameter optimization [37, 20, 9, 38], meta learning [9, 31, 22], neural network architecture search [30], reinforcement learning [15], and image processing [31]. In the paper, we consider solving the following nonconvex-strongly-convex bilevel optimization problem:

$$\min_{x \in \mathcal{X} \subseteq \mathbb{R}^{d_1}} f(x, y^*(x)) + h(x), \qquad \text{(Outer)} \qquad (1)$$

$$\text{s.t. } y^*(x) \in \arg \min_{y \in \mathbb{R}^{d_2}} g(x, y), \qquad \text{(Inner)}$$

where function $F(x) = f(x, y^*(x)) : \mathcal{X} \to \mathbb{R}$ is smooth and possibly nonconvex, and function $h(x)$ is convex and possibly nonsmooth, and function $g(x, y) : \mathcal{X} \times \mathbb{R}^{d_2} \to \mathbb{R}$ is $\mu$-strongly convex in $y \in \mathbb{R}^{d_2}$. The constraint set $\mathcal{X} \subseteq \mathbb{R}^{d_1}$ is compact and convex. Problem (1) covers a rich class of nonconvex objective functions with nonsmooth regularization, and is more general than the existing nonconvex bilevel optimization formulation in [11, 22] that does not consider any nonsmooth regularization. Here the function $h(x)$ can be the nonsmooth regularization term such as $h(x) = \lambda \|x\|_1$.

36th Conference on Neural Information Processing Systems (NeurIPS 2022).

Table 1: Comparisons of the representative bilevel optimization algorithms for finding an $\epsilon$-stationary point of the **deterministic** nonconvex-strongly-convex Problem (1) with $h(x)$ or without $h(x)$, *i.e.*, $\|\nabla F(x)\|^2 \leq \epsilon$ or its equivalent variants. $Gc(f,\epsilon)$ and $Gc(g,\epsilon)$ denote the number of gradient evaluations *w.r.t.* $f(x,y)$ and $g(x,y)$; $JV(g,\epsilon)$ denotes the number of Jacobian-vector products; $HV(g,\epsilon)$ is the number of Hessian-vector products; $\kappa = L/\mu$ is the conditional number. $\sqrt{}$ means that the algorithms solve both the **smooth** and **nonsmooth** bilevel optimizations.

| Algorithm | Reference | $Gc(f,\epsilon)$ | $Gc(g,\epsilon)$ | $JV(g,\epsilon)$ | $HV(g,\epsilon)$ | Nonsmooth |
|---|---|---|---|---|---|---|
| AID-BiO | [11] | $O(\kappa^4\epsilon^{-1})$ | $O(\kappa^5\epsilon^{-5/4})$ | $O(\kappa^4\epsilon^{-1})$ | $O(\kappa^{4.5}\epsilon^{-1})$ | |
| AID-BiO | [22] | $O(\kappa^3\epsilon^{-1})$ | $O(\kappa^4\epsilon^{-1})$ | $O(\kappa^3\epsilon^{-1})$ | $O(\kappa^{3.5}\epsilon^{-1})$ | |
| ITD-BiO | [22] | $O(\kappa^3\epsilon^{-1})$ | $\tilde{O}(\kappa^4\epsilon^{-1})$ | $\tilde{O}(\kappa^4\epsilon^{-1})$ | $\tilde{O}(\kappa^4\epsilon^{-1})$ | |
| BiO-BreD | Ours | $O(\kappa^2\epsilon^{-1})$ | $\tilde{O}(\kappa^3\epsilon^{-1})$ | $\tilde{O}(\kappa^3\epsilon^{-1})$ | $\tilde{O}(\kappa^3\epsilon^{-1})$ | $\sqrt{}$ |

Table 2: Comparisons of the representative bilevel optimization algorithms for finding an $\epsilon$-stationary point of the **stochastic** nonconvex-strongly-convex problem (2) with $h(x)$ or without $h(x)$, *i.e.*, $\mathbb{E}\|\nabla F(x)\|^2 \leq \epsilon$ or its equivalent variants. Since some algorithms do not provide the explicit dependence on $\kappa$, we use $p(\kappa)$.

| Algorithm | Reference | $Gc(f,\epsilon)$ | $Gc(g,\epsilon)$ | $JV(g,\epsilon)$ | $HV(g,\epsilon)$ | Nonsmooth |
|---|---|---|---|---|---|---|
| TTSA | [15] | $O(p(\kappa)\epsilon^{-2.5})$ | $O(p(\kappa)\epsilon^{-2.5})$ | $O(p(\kappa)\epsilon^{-2.5})$ | $O(p(\kappa)\epsilon^{-2.5})$ | |
| STABLE | [5] | $O(p(\kappa)\epsilon^{-2})$ | $O(p(\kappa)\epsilon^{-2})$ | $O(p(\kappa)\epsilon^{-2})$ | $O(p(\kappa)\epsilon^{-2})$ | |
| SMB | [13] | $O(p(\kappa)\epsilon^{-2})$ | $O(p(\kappa)\epsilon^{-2})$ | $O(p(\kappa)\epsilon^{-2})$ | $O(p(\kappa)\epsilon^{-2})$ | |
| VRBO | [41] | $O(p(\kappa)\epsilon^{-1.5})$ | $O(p(\kappa)\epsilon^{-1.5})$ | $O(p(\kappa)\epsilon^{-1.5})$ | $O(p(\kappa)\epsilon^{-1.5})$ | |
| SUSTAIN | [23] | $O(p(\kappa)\epsilon^{-1.5})$ | $O(p(\kappa)\epsilon^{-1.5})$ | $O(p(\kappa)\epsilon^{-1.5})$ | $O(p(\kappa)\epsilon^{-1.5})$ | |
| VR-saBiAdam | [18] | $O(p(\kappa)\epsilon^{-1.5})$ | $O(p(\kappa)\epsilon^{-1.5})$ | $O(p(\kappa)\epsilon^{-1.5})$ | $O(p(\kappa)\epsilon^{-1.5})$ | |
| BSA | [11] | $O(\kappa^6\epsilon^{-2})$ | $O(\kappa^9\epsilon^{-3})$ | $O(\kappa^6\epsilon^{-2})$ | $\tilde{O}(\kappa^6\epsilon^{-2})$ | |
| stocBiO | [22] | $O(\kappa^5\epsilon^{-2})$ | $O(\kappa^9\epsilon^{-2})$ | $O(\kappa^5\epsilon^{-2})$ | $\tilde{O}(\kappa^6\epsilon^{-2})$ | |
| SBiO-BreD | Ours | $O(\kappa^5\epsilon^{-2})$ | $O(\kappa^5\epsilon^{-2})$ | $O(\kappa^5\epsilon^{-2})$ | $\tilde{O}(\kappa^6\epsilon^{-2})$ | $\sqrt{}$ |
| ASBiO-BreD | Ours | $O(\kappa^5\epsilon^{-1.5})$ | $O(\kappa^5\epsilon^{-1.5})$ | $O(\kappa^5\epsilon^{-1.5})$ | $\tilde{O}(\kappa^6\epsilon^{-1.5})$ | $\sqrt{}$ |

Many recent machine learning research problems utilize the stochastic loss functions. Thus, we also consider the following stochastic bilevel optimization problem:

$$\min_{x\in\mathcal{X}\subseteq\mathbb{R}^{d_1}} \mathbb{E}_{\xi\sim\mathcal{D}}\big[f(x,y^*(x);\xi)\big] + h(x), \qquad \text{(Outer)} \qquad (2)$$

$$\text{s.t. } y^*(x) \in \arg\min_{y\in\mathbb{R}^{d_2}} \mathbb{E}_{\zeta\sim\mathcal{D}'}\big[g(x,y;\zeta)\big], \qquad \text{(Inner)}$$

where function $F(x) = \mathbb{E}_\xi\big[F(x;\xi)\big] = \mathbb{E}_\xi\big[f(x,y^*(x);\xi)\big]$ is smooth and possibly nonconvex, and function $h(x)$ is convex and possibly nonsmooth, and function $g(x,y) = \mathbb{E}_\zeta\big[g(x,y;\zeta)\big] : \mathcal{X}\times\mathbb{R}^{d_2} \to \mathbb{R}$ is $\mu$-strongly convex in $y \in \mathbb{R}^{d_2}$. $\xi$ and $\zeta$ are random variables following unknown distributions $\mathcal{D}$ and $\mathcal{D}'$, respectively. Both Problem (1) and Problem (2) have been used in many machine learning tasks with a hierarchical structure, such as hyper-parameter meta-learning [9, 22] and neural network architecture search [30].

Many bilevel optimization methods recently have been developed to solve these problems. For example, [11, 22] introduced a class of effective methods to solve the above deterministic Problem (1) and stochastic Problem (2) with $h(x) = 0$. However, these methods suffer from high computational complexity issue. More recently, multiple accelerated methods were designed for stochastic Problem (2) with $h(x) = 0$. Specifically, [5, 23, 14, 41] proposed accelerated bilevel optimization algorithms via using the variance reduced techniques of SARAH/SPIDER/SNVRG [36, 8, 40, 43] and STORM [6]. However, these accelerated methods obtain a lower computational complexity without considering the condition number, which also accounts for an important part of the computational complexity (please see Tables 1 and 2). Meanwhile, these accelerated methods only focus on the special case of the stochastic bilevel optimization Problem (2) with $h(x) = 0$.

To fill in the gaps, in the paper, we propose a class of efficient bilevel optimization methods with lower computational complexity to solve the bilevel optimization Problems (1) and (2), where the outer subproblem is nonconvex and possibly nonsmooth, and the inner subproblem is strongly convex. Specifically, we use the mirror decent iteration to update the variable $x$ based on the Bregman distance. Our main contributions are summarized as follows:

(i) We propose a class of enhanced bilevel optimization methods based on Bregman distance to solve the nonconvex-strongly-convex bilevel optimization problems. Moreover, we provide a comprehensive convergence analysis framework for our proposed methods.

(ii) An efficient bilevel optimization method based on Bregman distances (BiO-BreD) is proposed to solve the deterministic bilevel Problem (1). We prove that our BiO-BreD achieves a lower sample complexity than the best known results (please see Table 1).

(iii) We introduce an efficient bilevel optimization method based on adaptive Bregman distances (SBiO-BreD) to solve the stochastic bilevel Problem (2). Moreover, we design an accelerated version of SBiO-BreD algorithm (ASBiO-BreD) via using the variance reduced technique, which achieves a lower sample complexity than the best known results (please see Table 2).

**Note that** our methods can solve the constrained bilevel optimization with nonsmooth regularization but not rely on any form of constraint set and nonsmooth regularization. In the other words, our methods can solve the unconstrained bilevel optimization without nonsmooth regularization studied in [11, 22]. Naturally, our convergence analysis can be applied to both the constrained bilevel optimization with nonsmooth regularization and the unconstrained bilevel optimization without nonsmooth regularization.

## 2 Related Works

In this section, we will revisit the existing bilevel optimization algorithms and Bregman distance based methods.

### 2.1 Bilevel Optimization Methods

Bilevel optimization recently has attracted increasing interest in many machine learning applications such as model-agnostic meta-learning, neural network architecture search, and policy optimization. Thus, recently many algorithms [9, 11, 15, 34, 35, 22, 28] have been proposed to solve the bilevel optimization problems. Specifically, [11] proposed a class of approximation methods for bilevel optimization and studied convergence properties of the proposed methods under convexity assumption. [34, 35] developed the gradient-based descent aggregation methods for convex bilevel optimization. [37] presented a nonlinear primal–dual algorithm for nonsmooth convex bilevel optimization in parameter learning problems.

In parallel, [15] introduced a two-timescale stochastic algorithm framework for nonconvex stochastic bilevel optimization in reinforcement learning. Multiple accelerated bilevel approximation methods were developed later. Specifically, [22] proposed faster bilevel optimization methods based on the approximated implicit differentiation (AID) and iterative differentiation (ITD), respectively. [5, 23, 14, 41] presented several accelerated bilevel methods for the stochastic bilevel problems using variance-reduced techniques. More recently, [18] proposed a class of efficient adaptive methods for nonconvex-strongly-convex bilevel optimization problems. At the same time, the lower bound of bilevel optimization methods has been studied in [21] for these nonconvex-strongly-convex bilevel optimization problems. In addition, [34, 27, 32, 33] designed a class of value-function-based and gradient-based bilevel methods for nonconvex bilevel optimization problems and studied asymptotic convergence properties of these methods. [38] analyzed a class of special nonconvex nonsmooth bilevel optimization methods for selecting the best hyperparameter value for the nonsmooth $\ell_p$ regularization with $0 < p \leq 1$.

### 2.2 Bregman Distance-Based Methods

Bregman distance-based method (*a.k.a*, mirror descent method) [4, 1] is a powerful optimization tool because it uses the Bregman distance to fit the geometry of optimization problems. Bregman distance was first proposed in [2], and later extended in [3]. [4] introduced the first proximal minimization algorithm with Bregman function. [1] studied the mirror descent for convex optimization. [7] presented an effective variant of mirror descent, *i.e.* composite objective mirror descent, for regularized convex optimization. Subsequently, [42] studied the convergence properties of mirror descent algorithm for solving nonsmooth nonconvex problems. [26] integrated the variance reduced technique to the mirror descent algorithm for stochastic convex optimization. The variance-reduced adaptive stochastic mirror descent algorithm [29] was proposed to solve the nonsmooth nonconvex finite-sum optimization. More recently, [16] studied Bregman gradient methods for policy optimization.

## 3 Preliminaries

### 3.1 Notations

Let $I_d$ denote a $d$-dimensional identity matrix. $\mathcal{U}\{1, 2, \cdots, K\}$ denotes a uniform distribution over a discrete set $\{1, 2, \cdots, K\}$. $\|\cdot\|$ denotes the $\ell_2$-norm for vectors and spectral norm for matrices,

respectively. For two vectors $x$ and $y$, $\langle x, y \rangle$ denotes their inner product. $\nabla_x f(x, y)$ and $\nabla_y f(x, y)$ are the partial derivatives *w.r.t.* variables $x$ and $y$. Given the mini-batch samples $\mathcal{B} = \{\xi^i\}_{i=1}^b$, we define $\nabla f(x; \mathcal{B}) = \frac{1}{b} \sum_{i=1}^b \nabla f(x; \xi^i)$. For two sequences $\{a_n, b_n\}_{i=1}^n$, $a_n = O(b_n)$ denotes that $a_n \leq C b_n$ for some constant $C > 0$. The notation $\tilde{O}(\cdot)$ hides logarithmic terms. Given a convex closed set $\mathcal{X}$, we define a projection operation $\mathcal{P}_{\mathcal{X}}(x_0) = \arg\min_{x \in \mathcal{X}} \|x - x_0\|^2$. $\partial h(x)$ is the subgradient set of function $h(x)$.

### 3.2 Some Mild Assumptions

**Assumption 1.** *Function $F(x) = f(x, y^*(x))$ is possibly nonconvex w.r.t. $x$, and function $g(x, y)$ is $\mu$-strongly convex w.r.t. $y$. For stochastic case, the same assumptions hold for $f(x, y^*(x); \xi)$ and $g(x, y; \zeta)$, respectively.*

**Assumption 2.** *Functions $f(x, y)$ and $g(x, y)$ satisfy*

1) *$\|\nabla_y f(x, y)\| \leq C_{fy}$ and $\|\nabla_{xy}^2 g(x, y)\| \leq C_{gxy}$ for any $x \in \mathcal{X}$ and $y \in \mathbb{R}^{d_2}$;*

2) *The partial derivatives $\nabla_x f(x, y)$, $\nabla_y f(x, y)$, $\nabla_x g(x, y)$ and $\nabla_y g(x, y)$ are L-Lipschitz, e.g., for $x, x_1, x_2 \in \mathcal{X}$ and $y, y_1, y_2 \in \mathbb{R}^{d_2}$,*

$$\|\nabla_x f(x_1, y) - \nabla_x f(x_2, y)\| \leq L \|x_1 - x_2\|, \quad \|\nabla_x f(x, y_1) - \nabla_x f(x, y_2)\| \leq L \|y_1 - y_2\|.$$

*For stochastic case, the same assumptions hold for $f(x, y; \xi)$ and $g(x, y; \zeta)$ for any $\xi$ and $\zeta$.*

**Assumption 3.** *The partial derivatives $\nabla_{xy}^2 g(x, y)$ and $\nabla_{yy}^2 g(x, y)$ are $L_{gxy}$-Lipschitz and $L_{gyy}$-Lipschitz, e.g., for all $x, x_1, x_2 \in \mathcal{X}$ and $y, y_1, y_2 \in \mathbb{R}^{d_2}$*

$$\|\nabla_{xy}^2 g(x_1, y) - \nabla_{xy}^2 g(x_2, y)\| \leq L_{gxy} \|x_1 - x_2\|, \quad \|\nabla_{xy}^2 g(x, y_1) - \nabla_{xy}^2 g(x, y_2)\| \leq L_{gxy} \|y_1 - y_2\|.$$

*For stochastic case, the same assumptions hold for $\nabla_{xy}^2 g(x, y; \zeta)$ and $\nabla_y^2 g(x, y; \zeta)$ for any $\zeta$.*

**Assumption 4.** *Function $h(x)$ for any $x \in \mathcal{X}$ is convex but possibly nonsmooth.*

**Assumption 5.** *Function $\Phi(x) = F(x) + h(x)$ is bounded below, i.e., $\Phi^* = \inf_{x \in \mathcal{X}} \Phi(x) > -\infty$.*

Assumptions 1-3 are commonly used in bilevel optimization methods [11, 22, 23]. According to Assumption 1, $\|f(x, y_1) - f(x, y_2)\| = \|\nabla_y f(x, y_\tau)(y_1 - y_2)\| \leq \|\nabla_y f(x, y_\tau)\| \|y_1 - y_2\| \leq C_{fy} \|y_1 - y_2\|$, where $y_\tau = \tau y_1 + (1 - \tau) y_2$ and $\tau \in [0, 1]$. Thus $\|\nabla_y f(x, y)\| \leq C_{fy}$ is similar to the assumption that the function $f$ is $M$-Lipschitz in [22]. From the proofs in [22], we can find that they still use the norm bounded partial derivative $\|\nabla_y f(x, y)\| \leq M$. Similarly, according to Assumption 1, we have $\|\nabla_y g(x_1, y) - \nabla_y g(x_2, y)\| \leq L \|x_1 - x_2\|$. Since $\|\nabla_y g(x_1, y) - \nabla_y g(x_2, y)\| = \|\nabla_{xy}^2 g(x_{\tau'}, y)(x_1 - x_2)\| \leq \|\nabla_{xy}^2 g(x_{\tau'}, y)\| \|x_1 - x_2\| \leq C_{gxy} \|x_1 - x_2\|$, where $x_{\tau'} = \tau' x_1 + (1 - \tau') x_2$ and $\tau' \in [0, 1]$, we can let $C_{gxy} = L$ as in [22]. From the proofs in [22], we can find that they still use the norm bounded partial derivative $\|\nabla_{xy}^2 g(x, y)\| \leq L$ for all $x, y$. Throughout the paper, we let $C_{gxy} = L$. Assumption 4 is generally used for regularization such as $h(x) = \|x\|_1$. Assumption 5 ensures the feasibility of Problems (1) and (2).

When we use the first-order methods to solve the above bilevel optimization Problems (1) and (2), we can easily obtain the partial (stochastic) derivative $\nabla_y g(x, y)$ or $\nabla_y g(x, y; \zeta)$ to update variable $y$. However, it is hard to get the (stochastic) gradient $\nabla F(x) = \frac{\partial f(x, y^*(x))}{\partial x}$ or $\nabla F(x; \xi) = \frac{\partial f(x, y^*(x); \xi)}{\partial x}$, when there is no closed form solution for the inner problem of Problems (1) and (2). Thus, a key point of solving the Problems (1) and (2) is to estimate the gradient $\nabla F(x)$. The following lemma provides one gradient estimator of $\nabla F(x)$.

**Lemma 1.** *(Lemma 2.1 in [11]) Under the above Assumptions (1, 2, 3), we have, for any $x \in \mathcal{X}$*

$$\begin{aligned} \nabla F(x) &= \nabla_x f(x, y^*(x)) + \nabla y^*(x)^T \nabla_y f(x, y^*(x)) \\ &= \nabla_x f(x, y^*(x)) - \nabla_{xy}^2 g(x, y^*(x)) [\nabla_{yy}^2 g(x, y^*(x))]^{-1} \nabla_y f(x, y^*(x)). \end{aligned} \tag{3}$$

Lemma 1 provides a natural estimator of $\nabla F(x)$, defined as, for all $x \in \mathcal{X}, y \in \mathbb{R}^{d_2}$

$$\bar{\nabla} f(x, y) = \nabla_x f(x, y) - \nabla_{xy}^2 g(x, y) \left( \nabla_{yy}^2 g(x, y) \right)^{-1} \nabla_y f(x, y). \tag{4}$$

Next, we show some properties of $\nabla F(x)$, $y^*(x)$ and $\bar{\nabla} f(x, y)$ in the following lemma:

---

**Algorithm 1** Deterministic BiO-BreD Algorithm

---
1: **Input:** $T$, $K \geq 1$, learning rates $\gamma > 0$, $\lambda > 0$;
2: **initialize:** $x_0 \in \mathcal{X}$ and $y_{-1}^K = y_0 \in \mathbb{R}^{d_2}$;
3: **for** $t = 0, 1, \cdots, T-1$ **do**
4:     Let $y_t^0 = y_{t-1}^K$;
5:     **for** $k = 1, \cdots, K$ **do**
6:        Update $y_t^k = y_t^{k-1} - \lambda \nabla_y g(x_t, y_t^{k-1})$;
7:     **end for**
8:     Compute partial derivative $w_t = \frac{\partial f(x_t, y_t^K)}{\partial x}$ via backpropagation *w.r.t.* $x_t$;
9:     Given a $\rho$-strongly convex mirror function $\psi_t$;
10:     Update $x_{t+1} = \arg\min_{x \in \mathcal{X}} \left\{ \langle w_t, x \rangle + h(x) + \frac{1}{\gamma} D_{\psi_t}(x, x_t) \right\}$;
11: **end for**
12: **Output:** Uniformly and randomly choose from $\{x_t, y_t\}_{t=1}^T$.

---

**Lemma 2.** *(Lemma 2.2 in [11]) Under the Assumptions (1, 2, 3), for all $x, x_1, x_2 \in \mathcal{X}$ and $y \in \mathbb{R}^{d_2}$, we have $\|\bar{\nabla} f(x, y) - \nabla F(x)\| \leq L_y \|y^*(x) - y\|$*

$$\|y^*(x_1) - y^*(x_2)\| \leq \kappa \|x_1 - x_2\|, \quad \|\nabla F(x_1) - \nabla F(x_2)\| \leq L_F \|x_1 - x_2\|,$$

*where $L_y = L + \frac{L^2}{\mu} + \frac{C_{fy} L_{gxy}}{\mu} + \frac{L_{gyy} C_{fy} L}{\mu^2}$, $\kappa = \frac{L}{\mu}$, and $L_F = L + \frac{2L^2 + L_{gxy} C_{fy}^2}{\mu} + \frac{L_{gyy} C_{fy} L + L^3 + L_{gxy} C_{fy} L}{\mu^2} + \frac{L_{gyy} C_{fy} L^2}{\mu^3}$.*

# 4 Bilevel Optimization via Bregman Distance Methods

In this section, we propose a class of enhanced bilevel optimization methods based on Bregman distance to solve the deterministic Problem (1) and the stochastic Problem (2), respectively.

## 4.1 Deterministic BiO-BreD Algorithm

In this subsection, we propose an efficient deterministic bilevel optimization method via Bregman distances (BiO-BreD) to solve the deterministic Problem (1). Algorithm 1 summarizes the algorithmic framework of our BiO-BreD method.

Given a $\rho$-strongly convex and continuously-differentiable function $\psi(x)$, *i.e.*, $\langle x_1 - x_2, \nabla\psi(x_1) - \nabla\psi(x_2)\rangle \geq \rho \|x_1 - x_2\|^2$, we define a Bregman distance [3, 4] for any $x_1, x_2 \in \mathcal{X}$:

$$D_\psi(x_1, x_2) = \psi(x_1) - \psi(x_2) - \langle \nabla\psi(x_2), x_1 - x_2 \rangle.$$

In Algorithm 1, we use the mirror descent iteration to update the variable $x$ at $t+1$-th step:

$$x_{t+1} = \arg\min_{x \in \mathcal{X}} \left\{ \langle w_t, x \rangle + h(x) + \frac{1}{\gamma} D_{\psi_t}(x, x_t) \right\}, \tag{5}$$

where $\gamma > 0$ is stepsize, and $w_t$ is an estimator of $\nabla F(x_t)$. Here the mirror function $\psi_t$ can be dynamic as the algorithm is running. Let $\psi_t(x) = \frac{1}{2}\|x\|^2$, we have $D_{\psi_t}(x, x_t) = \frac{1}{2}\|x - x_t\|^2$. When $\mathcal{X} = \mathbb{R}^{d_1}$, the above subproblem (5) is equivalent to the proximal gradient descent. When $\mathcal{X} \subseteq \mathbb{R}^{d_1}$ and $h(x) = 0$, the above subproblem (5) is equivalent to the projection gradient descent. Let $\psi_t(x) = \frac{1}{2}x^T H_t x$, we have $D_{\psi_t}(x, x_t) = \frac{1}{2}(x - x_t)^T H_t(x - x_t)$. When $H_t$ is an approximated Hessian matrix, the above subproblem (5) is equivalent to the proximal quasi-Newton decent. When $H_t$ is an adaptive matrix as used in [19], the above subproblem (5) is equivalent to the proximal adaptive gradient decent.

In Algorithm 1, we use gradient estimator $w_t = \frac{\partial f(x_t, y_t^K)}{\partial x}$ to estimate $\nabla F(x_t)$, where the partial derivative $w_t = \frac{\partial f(x_t, y_t^K)}{\partial x}$ is obtained by the backpropagation *w.r.t.* $x_t$.

## 4.2 SBiO-BreD Algorithm

In this subsection, we introduce an efficient stochastic bilevel optimization method via Bregman distance (SBiO-BreD) to solve the stochastic bilevel optimization Problem (2). Algorithm 2 describes the algorithmic framework of our SBiO-BreD method.

---

**Algorithm 2** Stochastic BiO-BreD (SBiO-BreD) Algorithm

---

1: **Input:** $T, K \geq 1$, stepsizes $\gamma > 0$, $\lambda > 0$, $\{\eta_t\}_{t=1}^T$;
2: **initialize:** $x_0 \in \mathcal{X}$ and $y_0 \in \mathbb{R}^{d_2}$;
3: **for** $t = 0, 1, \cdots, T-1$ **do**
4:    Draw randomly $b$ independent samples $\mathcal{B}_t = \{\zeta_t^i\}_{i=1}^b$, and compute stochastic partial deriva-
       tives $v_t = \nabla_y g(x_t, y_t; \mathcal{B}_t)$;
5:    Update $y_{t+1} = y_t - \lambda \eta_t v_t$;
6:    Draw randomly $b(K+1)$ independent samples $\bar{\mathcal{B}}_t = \{\xi_{t,i}, \zeta_{t,i}^0 \cdots, \zeta_{t,i}^{K-1}\}_{i=1}^b$, and compute
       stochastic partial derivatives $w_t = \bar{\nabla} f(x_t, y_t; \bar{\mathcal{B}}_t)$;
7:    Given a $\rho$-strongly convex mirror function $\psi_t$;
8:    Update $x_{t+1} = \arg\min_{x \in \mathcal{X}} \left\{ \langle w_t, x \rangle + h(x) + \frac{1}{\gamma} D_{\psi_t}(x, x_t) \right\}$;
9: **end for**
10: **Output:** Uniformly and randomly choose from $\{x_t, y_t\}_{t=1}^T$.

---

Given $K \geq 1$ and draw $K+1$ independent samples $\bar{\xi} = \{\xi, \zeta^0, \cdots, \zeta^{K-1}\}$, as in [15, 23], we definite a stochastic gradient estimator:

$$\bar{\nabla} f(x, y, \bar{\xi}) = \nabla_x f(x, y; \xi) - \nabla_{xy}^2 g(x, y; \zeta^0) \left[ \frac{K}{L} \prod_{i=1}^k \left( I_{d_2} - \frac{1}{L} \nabla_{yy}^2 g(x, y; \zeta^i) \right) \right] \nabla_y f(x, y; \xi), \quad (6)$$

where $k \sim \mathcal{U}\{0, 1, \cdots, K-1\}$ is a uniform random variable independent on $\bar{\xi}$. It is easy to verify that $\bar{\nabla} f(x, y, \bar{\xi})$ is a biased estimator of $\bar{\nabla} f(x, y)$, *i.e.* $\mathbb{E}_{\bar{\xi}}[\bar{\nabla} f(x, y; \bar{\xi})] \neq \bar{\nabla} f(x, y)$. For the gradient estimator (6), thus we define a bias $R(x, y) = \bar{\nabla} f(x, y) - \mathbb{E}_{\bar{\xi}}[\bar{\nabla} f(x, y; \bar{\xi})] : \mathcal{X} \times \mathbb{R}^{d_2} \to \mathbb{R}$.

**Lemma 3.** *( Lemma 2.1 in [23] ) Under the about Assumptions (1, 2, 3), for any $K \geq 1$, the gradient estimator in (6) satisfies*

$$\|R(x, y)\| \leq \frac{LC_{fy}}{\mu} \left( 1 - \frac{\mu}{L} \right)^K.$$

Lemma 5 shows that the bias $R(x, y)$ decays exponentially fast with number $K$, and with choosing $K = \frac{L}{\mu} \log(LC_{fy}T/\mu)$, we have $\|R(x, y)\| \leq \frac{1}{T}$. Let $\frac{LC_{fy}}{\mu} \left( 1 - \frac{\mu}{L} \right)^K \leq \frac{1}{T}$, we have $K \log(1 - \frac{\mu}{L}) \leq \log(\frac{\mu}{LC_{fy}T})$. Due to $\mu < L$, we have $K \geq \log(\frac{C_{fy}LT}{\mu}) / \log(\frac{L}{L-\mu})$. Further due to $\frac{\mu}{L} \leq \log(\frac{L}{L-\mu})$, let $K = \frac{L}{\mu} \log(LC_{fy}T/\mu)$, we have $\|R(x, y)\| \leq \frac{1}{T}$. Note that here we use $C_{gxy} = L$.

To simplify notations, let $\bar{\xi}_t^i = \{\xi_{t,i}, \zeta_{t,i}^0 \cdots, \zeta_{t,i}^{K-1}\}$. In Algorithm 2, we use mini-batch stochastic gradient estimator $w_t = \bar{\nabla} f(x_t, y_t; \bar{\mathcal{B}}_t) = \frac{1}{b} \sum_{i=1}^b \bar{\nabla} f(x_t, y_t; \bar{\xi}_t^i)$, where $\bar{\nabla} f(x_t, y_t; \bar{\xi}_t^i)$

$$= \nabla_x f(x_t, y_t; \xi_{t,i}) - \nabla_{xy}^2 g(x_t, y_t; \zeta_{t,i}^0) \left[ \frac{K}{L} \prod_{j=1}^k \left( I_{d_2} - \frac{1}{L} \nabla_{yy}^2 g(x_t, y_t; \zeta_{t,i}^j) \right) \right] \nabla_y f(x_t, y_t; \xi_{t,i}),$$

with $k \sim \mathcal{U}\{0, 1, \cdots, K-1\}$. Let $R(x_t, y_t) = w_t - \bar{\nabla} f(x_t, y_t) = \bar{\nabla} f(x_t, y_t; \bar{\mathcal{B}}_t) - \bar{\nabla} f(x_t, y_t)$, we have $\mathbb{E}[\bar{\nabla} f(x_t, y_t; \bar{\mathcal{B}}_t)] = R(x_t, y_t) + \bar{\nabla} f(x_t, y_t)$. According to the above Lemma 5, it is easy to verify that $\|R(x_t, y_t)\| \leq \frac{LC_{fy}}{\mu} \left( 1 - \frac{\mu}{L} \right)^K$.

### 4.3 ASBiO-BreD Algorithm

In this subsection, we propose an accelerated version of SBiO-BreD method (ASBiO-BreD) to solve the stochastic bilevel optimization Problem (2) via using variance reduced technique of SARAH/SPIDER/SNVRG [36, 8, 40, 43]. Algorithm 3 shows the algorithmic framework of the ASBiO-BreD method.

In Algorithm 3, we use the variance reduced technique of SPIDER to accelerate SBiO-BreD algorithm. When $\mod (t, q) = 0$, we draw a relative large batch samples $\mathcal{B}_t = \{\zeta_t^i\}_{i=1}^b$ and $\bar{\mathcal{B}}_t = \{\bar{\xi}_t^i\}_{i=1}^b$ to estimate our stochastic partial derivatives $v_t$ and $w_t$, respectively. When $\mod (t, q) \neq 0$, we draw a mini-batch samples $\mathcal{I}_t = \{\xi_t^i\}_{i=1}^{b_1}$ and $\bar{\mathcal{I}}_t = \{\bar{\xi}_t^i\}_{i=1}^{b_1}$ $(b > b_1)$ to estimate $v_t$ and $w_t$, respectively. Let $R(x_t, y_t) = \bar{\nabla} f(x_t, y_t; \bar{\mathcal{I}}_t) - \bar{\nabla} f(x_t, y_t)$ when $\mod (t, q) \neq 0$, we have $\mathbb{E}[\bar{\nabla} f(x_t, y_t; \bar{\mathcal{I}}_t)] = R(x_t, y_t) + \bar{\nabla} f(x_t, y_t)$ and $\|R(x_t, y_t)\| \leq \frac{LC_{fy}}{\mu} \left( 1 - \frac{\mu}{L} \right)^K$.

---

**Algorithm 3** Accelerated Stochastic BiO-BreD (ASBiO-BreD) Algorithm

---

1: **Input:** $T, K \geq 1$, $q$, stepsizes $\gamma > 0$, $\lambda > 0$, $\{\eta_t\}_{t=1}^T$, mini-batch sizes $b$ and $b_1$;
2: **initialize:** $x_0 \in \mathcal{X}$ and $y_0 \in \mathbb{R}^{d_2}$;
3: **for** $t = 0, 1, \cdots, T-1$ **do**
4:    **if** $\mathrm{mod}\ (t, q) = 0$ **then**
5:       Draw randomly $b$ independent samples $\mathcal{B}_t = \{\zeta_t^i\}_{i=1}^b$, and compute stochastic partial derivative $v_t = \nabla_y g(x_t, y_t; \mathcal{B}_t)$;
6:       Draw randomly $b(K+1)$ independent samples $\bar{\mathcal{B}}_t = \{\xi_{t,i}, \zeta_{t,i}^0 \cdots, \zeta_{t,i}^{K-1}\}_{i=1}^b$, and compute stochastic partial derivative $w_t = \bar{\nabla} f(x_t, y_t; \bar{\mathcal{B}}_t)$;
7:    **else**
8:       Generate randomly $b_1$ independent samples $\mathcal{I}_t = \{\zeta_t^i\}_{i=1}^{b_1}$, and compute stochastic partial derivative $v_t = \nabla_y g(x_t, y_t; \mathcal{I}_t) - \nabla_y g(x_{t-1}, y_{t-1}; \mathcal{I}_t) + v_{t-1}$;
9:       Generate randomly $b_1(K+1)$ independent samples $\bar{\mathcal{I}}_t = \{\xi_{t,i}, \zeta_{t,i}^0 \cdots, \zeta_{t,i}^{K-1}\}_{i=1}^{b_1}$, and compute stochastic partial derivative $w_t = \bar{\nabla} f(x_t, y_t; \bar{\mathcal{I}}_t) - \bar{\nabla} f(x_{t-1}, y_{t-1}; \bar{\mathcal{I}}_t) + w_{t-1}$;
10:    **end if**
11:    Update $y_{t+1} = y_t - \lambda \eta_t v_t$;
12:    Given a $\rho$-strongly convex mirror function $\psi_t$;
13:    Update $x_{t+1} = \arg\min_{x \in \mathcal{X}} \left\{ \langle w_t, x \rangle + h(x) + \frac{1}{\gamma} D_{\psi_t}(x, x_t) \right\}$;
14: **end for**
15: **Output:** Uniformly and randomly choose from $\{x_t, y_t\}_{t=1}^T$.

---

## 5 Convergence Analysis

In this section, we study the convergence properties of our new algorithms (*i.e.*, BiO-BreD, SBiO-BreD, and ASBiO-BreD) under mild conditions. All proofs are provided in the Appendix B.

We begin with introducing a useful convergence metric $\|\mathcal{G}_t\|^2$ or $\mathbb{E}\|\mathcal{G}_t\|^2$ to measure convergence properties of our algorithms. Given the generated parameter vector $x_t$ at the $t$-th iteration in our algorithms, as in [10, 29], we define the generalized gradient at the $t$-th iteration as:

$$\mathcal{G}_t = \frac{1}{\gamma}(x_t - x_{t+1}^+), \quad x_{t+1}^+ = \arg\min_{x \in \mathcal{X}} \left\{ \langle \nabla F(x_t), x \rangle + h(x) + \frac{1}{\gamma} D_{\psi_t}(x, x_t) \right\},$$

where $F(x) = f(x, y^*(x))$ or $F(x) = \mathbb{E}_\xi[f(x, y^*(x); \xi)]$. When $\psi_t(x) = \frac{1}{2}\|x\|^2$, $\mathcal{X} = \mathbb{R}^{d_1}$ and $h(x) = c$ is a constant, we have $\|\mathcal{G}_t\|^2 = \|\nabla F(x_t)\|^2$, which is a common convergence metric used in [11, 22]. When $\psi(x) = \frac{1}{2}\|x\|^2$, $\mathcal{X} \subseteq \mathbb{R}^{d_1}$ and $h(x) = c$ is a constant, our convergence metric is $\|\mathcal{G}_t\|^2 = \|\frac{1}{\gamma}(x_t - \mathcal{P}_\mathcal{X}(x_t - \gamma \nabla F(x_t))\|^2$, which was also used in [15].

Next, we provide some useful lemmas and some mild assumptions.

**Lemma 4.** *(Lemma 3.1 in [23]) Under the above Assumptions (1, 2, 3), stochastic gradient estimator $\bar{\nabla} f(x, y; \bar{\xi})$ is $L_K$-Lipschitz continuous, e.g., for $x_1, x_2 \in \mathcal{X}$ and $y \in \mathbb{R}^{d_2}$,*

$$\mathbb{E}_{\bar{\xi}} \|\bar{\nabla} f(x_1, y; \bar{\xi}) - \bar{\nabla} f(x_2, y; \bar{\xi})\|^2 \leq L_K^2 \|x_1 - x_2\|^2,$$

*where $L_K^2 = 2L^2 + 6L^4 \frac{K}{2\mu L - \mu^2} + 6C_{fy}^2 L_{gxy}^2 \frac{K}{2\mu L - \mu^2} + 6L^4 \frac{K^3 L_{gyy}^2}{(L-\mu)^2(2\mu L - \mu^2)}$.*

**Lemma 5.** *Suppose the sequence $\{x_t, y_t\}_{t=1}^T$ be generated from Algorithms 2 and 3. Under the above assumptions, given $0 < \eta_t \leq 1$ for all $t \geq 1$ and $0 < \lambda \leq \frac{1}{6L}$, we have*

$$\|y_{t+1} - y^*(x_{t+1})\|^2 \leq (1 - \frac{\eta_t \mu \lambda}{4})\|y_t - y^*(x_t)\|^2 - \frac{3\eta_t \lambda^2}{4}\|v_t\|^2$$
$$+ \frac{25\eta_t \lambda}{6\mu}\|\nabla_y g(x_t, y_t) - v_t\|^2 + \frac{25\kappa^2}{6\eta_t \mu \lambda}\|x_{t+1} - x_t\|^2.$$

The above lemma 5 basically follows the Lemma 28 of [17] used for minimax optimization.

**Assumption 6.** *The stochastic partial derivative $\nabla_y g(x, y; \zeta)$ satisfies $\mathbb{E}[\nabla_y g(x, y; \zeta)] = \nabla_y g(x, y)$ and $\mathbb{E}\|\nabla_y g(x, y; \zeta) - \nabla_y g(x, y)\|^2 \leq \sigma^2$. The estimated stochastic partial derivative $\bar{\nabla} f(x, y; \bar{\xi})$ defined in (6) satisfies $\mathbb{E}_{\bar{\xi}}[\bar{\nabla} f(x, y; \bar{\xi})] = \bar{\nabla} f(x, y) + R(x, y)$ and $\mathbb{E}_{\bar{\xi}}\|\bar{\nabla} f(x, y; \bar{\xi}) - \bar{\nabla} f(x, y) - R(x, y)\|^2 \leq \sigma^2$.*

**Assumption 7.** *The mirror functions $\{\psi_t(x)\}_{t=0}^T$ are $\rho$-strongly convex, where $\rho > 0$.*

Assumption 6 is commonly used in stochastic bilevel optimization methods [15, 23]. Assumption 7 shows that the constant $\rho$ can be seen as a lower bound of the strong convexity of all the mirror functions $\psi_t(x)$ for all $t \geq 0$, which is widely used in mirror descent algorithms [29] and adaptive gradient algorithms [19].

## 5.1 Convergence Analysis of BiO-BreD Algorithm

In this subsection, we provide the convergence properties of our BiO-BreD algorithm.

**Theorem 1.** *Suppose the sequence $\{x_t, y_t\}_{t=1}^T$ be generated from Algorithm 1. Let $0 < \gamma \leq \frac{3\rho}{4L_F}$, $0 < \lambda < \frac{1}{L}$, $K = \log(T)/\log(\frac{1}{1-\lambda\mu}) + 1$ and $\|y_t^0 - y^*(x_t)\|^2 \leq \Delta$ for all $t \geq 0$, we have*

$$\frac{1}{T}\sum_{t=0}^{T-1}\|\mathcal{G}_t\|^2 \leq \frac{16\big(\Phi(x_0) - \Phi^*\big)}{3T\gamma\rho} + \frac{22\Delta L_1^2}{\rho^2 T} + \frac{22\Delta L_2^2}{\rho^2 T} + \frac{22L_3^2}{\rho^2 T^2}, \tag{7}$$

*where $\kappa = \frac{L}{\mu}$, $L_1 = \frac{L(L+\mu)}{\mu}$, $L_2 = \frac{2C_{fy}(\mu L_{gxy} + LL_{gyy})}{\mu^2}$ and $L_3 = \frac{LC_{fy}}{\mu}$.*

**Remark 1.** *Without loss of generality, let $L \geq \frac{1}{\mu}$, $\lambda = \frac{1}{2L}$, $\gamma = \frac{3\rho}{4L_F}$ and $\rho = O(L)$. It is easy to verify that our BiO-BreD algorithm has a convergence rate of $O\big(\frac{\kappa^2}{T}\big)$. Let $\frac{\kappa^2}{T} = \epsilon$, we have $T = \kappa^2\epsilon^{-1}$. Due to $K = \log(T)/\log(\frac{1}{1-\lambda\mu}) + 1$, we choose $K = O(\kappa \log(\frac{1}{\epsilon}))$ for finding $\epsilon$-stationary point of the problem (1), we need the gradient complexity: $Gc(f, \epsilon) = 2T = O(\kappa^2\epsilon^{-1})$ and $Gc(g, \epsilon) = KT = \tilde{O}(\kappa^3\epsilon^{-1})$, and the Jacobian-vector and Hessian-vector product complexities: $JV(g, \epsilon) = KT = \tilde{O}(\kappa^3\epsilon^{-1})$ and $HV(g, \epsilon) = KT = \tilde{O}(\kappa^3\epsilon^{-1})$.*

## 5.2 Convergence Analysis of SBiO-BreD Algorithm

In this subsection, we provide the convergence properties of our SBiO-BreD algorithm.

**Theorem 2.** *Suppose the sequence $\{x_t, y_t\}_{t=1}^T$ be generated from Algorithm 2. Let $\Delta = \|y_0 - y^*(x_0)\|^2$, $K = \frac{L}{\mu}\log(\frac{LC_{fy}T}{\mu})$, $0 < \eta = \eta_t \leq 1$, $0 < \gamma \leq \min\big(\frac{3\rho}{4L_F}, \frac{9\eta\rho\mu\lambda}{800\kappa^2}, \frac{\eta\mu\rho\lambda}{47L_y^2}\big)$ and $0 < \lambda \leq \frac{1}{6L}$, we have*

$$\frac{1}{T}\sum_{t=1}^T \mathbb{E}\|\mathcal{G}_t\|^2 \leq \frac{32(\Phi(x_0) - \Phi^*)}{3T\gamma\rho} + \frac{32\Delta}{3T\gamma\rho} + \frac{752\sigma^2}{3\rho^2 b} + \frac{400\eta\lambda\sigma^2}{9\gamma\rho\mu b} + \frac{752}{3\rho^2 T^2}. \tag{8}$$

**Remark 2.** *Without loss of generality, let $L \geq \frac{1}{\mu}$, $\lambda = \frac{1}{6L}$, $\gamma = \min\big(\frac{3\rho}{4L_F}, \frac{9\eta\rho\mu\lambda}{800\kappa^2}, \frac{\eta\mu\rho\lambda}{47L_y^2}\big)$ and $\rho = O(L)$, we have $\gamma\rho = O(\frac{1}{\kappa^3})$ It is easily verified that our SBiO-BreD algorithm has a convergence rate of $O\big(\frac{\kappa^3}{T} + \frac{\kappa^2}{b}\big)$. Let $\frac{\kappa^3}{T} = \frac{\epsilon}{2}$ and $\frac{\kappa^2}{b} = \frac{\epsilon}{2}$, we have $T = 2\kappa^3\epsilon^{-1}$ and $b = 2\kappa^2\epsilon^{-1}$. Due to $K = \frac{L}{\mu}\log(\frac{LC_{fy}T}{\mu})$, we have $K = O(\kappa\log(\frac{\kappa^4}{\epsilon})) = \tilde{O}(\kappa)$. For finding $\epsilon$-stationary point of the problem (2), we need the gradient complexity: $Gc(f, \epsilon) = 2bT = \kappa^5\epsilon^{-2}$ and $Gc(g, \epsilon) = bT = O(\kappa^5\epsilon^{-2})$, and the Jacobian-vector and Hessian-vector product complexities: $JV(g, \epsilon) = bT = O(\kappa^5\epsilon^{-2})$ and $HV(g, \epsilon) = KbT = \tilde{O}(\kappa^6\epsilon^{-2})$.*

## 5.3 Convergence Analysis of ASBiO-BreD Algorithm

In this subsection, we provide the convergence properties of our ASBiO-BreD algorithm.

**Theorem 3.** *Suppose the sequence $\{x_t, y_t\}_{t=1}^T$ be generated from Algorithm 3. Let $\Delta = \|y_0 - y^*(x_0)\|^2$, $b_1 = q$, $K = \frac{L}{\mu}\log(\frac{LC_{fy}T}{\mu})$, $0 < \eta = \eta_t \leq 1$, $0 < \gamma \leq \min\big(\frac{3\rho}{38L_K^2\eta}, \frac{3\rho}{4L_F}, \frac{2\rho\eta\mu\lambda}{19L_y^2}, \frac{\rho\eta}{8}, \frac{9\rho\eta\mu\lambda}{400\kappa^2}\big)$ and $0 < \lambda \leq \min\big(\frac{1}{6L}, \frac{9\mu}{100\eta^2 L^2}\big)$, we have*

$$\frac{1}{T}\sum_{t=0}^{T-1}\mathbb{E}\|\mathcal{G}_t\|^2 \leq \frac{32(\Phi(x_0) - \Phi^*)}{3T\gamma\rho} + \frac{32\Delta}{3T\gamma\rho} + \frac{152}{3T^2\rho^2} + \frac{4}{\eta\rho\gamma}\Big(\frac{1}{L^2} + \frac{1}{L_K^2}\Big)\frac{\sigma^2}{b}. \tag{9}$$

**Remark 3.** *Without loss of generality, let $L \geq \frac{1}{\mu}$, $\lambda = \min\big(\frac{1}{6L}, \frac{9\mu}{100\eta^2 L^2}\big)$, $\gamma = \min\big(\frac{3\rho}{38L_K^2\eta}, \frac{3\rho}{4L_F}, \frac{2\rho\eta\mu\lambda}{19L_y^2}, \frac{\rho\eta}{8}, \frac{9\rho\eta\mu\lambda}{400\kappa^2}\big)$ and $\rho = O(L)$, we have $\gamma\rho = O(\frac{1}{\kappa^4})$. It is easily verified that our ASBiO-BreD algorithm has a convergence rate of $O\big(\frac{\kappa^4}{T} + \frac{\kappa^2}{b}\big)$. Let $\frac{\kappa^4}{T} = \frac{\epsilon}{2}$ and $\frac{\kappa^2}{b} = \frac{\epsilon}{2}$, we have*

$T = 2\kappa^4\epsilon^{-1}$ and $b = 2\kappa^2\epsilon^{-1}$. Due to $K = \frac{L}{\mu}\log(\frac{LC_{f_y}T}{\mu})$, we have $K = O(\kappa\log(\frac{\kappa^4}{\epsilon})) = \tilde{O}(\kappa)$. Let $b_1 = q = \kappa\epsilon^{-0.5}$. For finding $\epsilon$-stationary point of the problem (2), we need the gradient complexity: $Gc(f,\epsilon) = 2(\frac{bT}{q} + 2b_1T) = O(\kappa^5\epsilon^{-1.5})$ and $Gc(g,\epsilon) = \frac{bT}{q} + 2b_1T = O(\kappa^5\epsilon^{-1.5})$, and the Jacobian-vector and Hessian-vector product complexities: $JV(g,\epsilon) = \frac{bT}{q} + 2b_1T = O(\kappa^5\epsilon^{-1.5})$ and $HV(g,\epsilon) = K(\frac{bT}{q} + 2b_1T) = \tilde{O}(\kappa^6\epsilon^{-1.5})$.

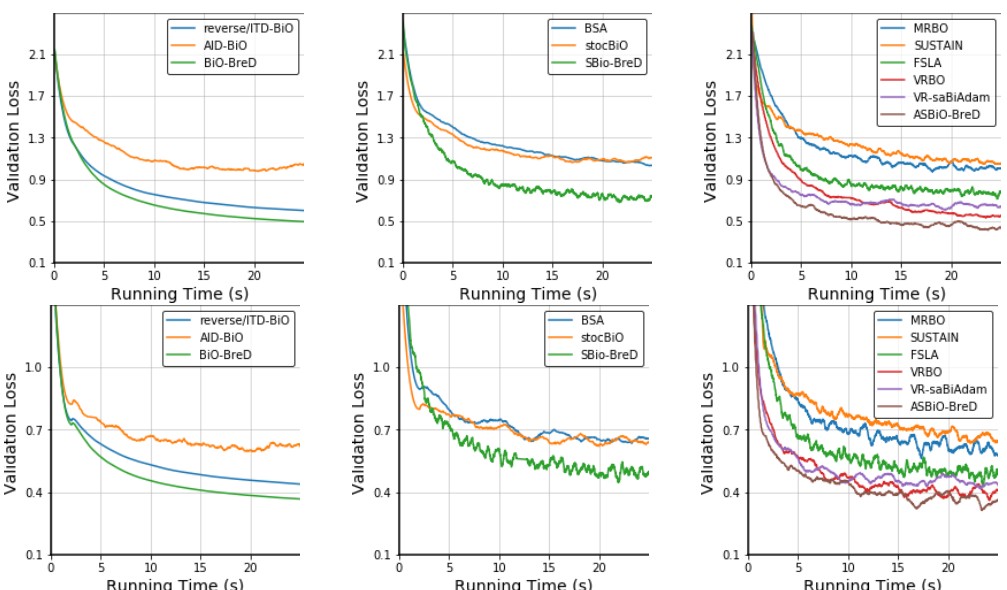

Figure 1: Validation Loss *vs*. Running Time for different methods. We compare our BiO-BreD with deterministic baselines (the first column), SBiO-BreD with stochastic baselines (the second column); ASBiO-BreD with momentum-based or SPIDER/SARAH based baselines (the last column). We test two values of $\varrho$: large noise setting $\varrho = 0.8$ (top row) and small noise setting $\varrho = 0.4$ (bottom row).

## 6  Numerical Experiments

In this section, we perform two tasks to demonstrate the efficiency of our algorithms: 1) data hyper-cleaning task [39] over the MNIST dataset [25]; 2) hyper-representation learning task [9] over the Omniglot dataset [24]. In the experiment, we compare our algorithms (*i.e.*, BiO-BreD, SBiO-BreD, and ASBiO-BreD) with the following bilevel optimization algorithms: reverse [9]/AID-BiO [11, 22], AID-CG [12], AID-FP [12], stocBiO [22]), MRBO [21], VRBO [21], FSLA [28], SUSTAIN [23], and VR-saBiAdam [18]. All experiments are averaged over 5 runs and we use a server with AMD EPYC 7763 64-Core CPU and 1 NVIDIA RTX A5000.

We use Bregman function $\psi_t(x) = \frac{1}{2}x^T H_t x$ to generate the Bregman distance in our algorithms, where $H_t$ is the adaptive matrix as used in [19], *i.e.* the exponential moving average of the square of the gradient and we use coefficient 0.99 in all experiments.

### 6.1  Data Hyper-cleaning

In this subsection, we perform data hyper-cleaning over the MNIST dataset [25]. The formulation of this problem is as follows:

$$\min_\lambda l_{val}(\lambda, w^*(\lambda)) := \frac{1}{|D_\mathcal{V}|}\sum_{(x_i,y_i)\in D_\mathcal{V}} l(x_i^T w^*(\lambda), y_i)$$

$$\text{s.t. } w^*(\lambda) = \arg\min_w l_{tr}(\lambda, w) := \frac{1}{|D_\mathcal{T}|}\sum_{(x_i,y_i)\in D_\mathcal{T}} \sigma(\lambda_i)l(x_i^T w, y_i) + C\|w\|^2,$$

where $l(\cdot)$ denotes the cross entropy loss, $D_\mathcal{T}$ and $D_\mathcal{V}$ are training and validation datasets, respectively. Here $\lambda = \{\lambda_i\}_{i\in D_\mathcal{T}}$ are hyper-parameters and $C \geq 0$ is a tuning parameter, $\sigma(\cdot)$ denotes the sigmoid function. In experiment, we set $C = 0.001$. The dataset includes a training set and a validation set

Table 3: Validation accuracy *vs.* Running Time (5-way-1-shot) for different methods (with $L_1$ regularization)

| Time | AID_BiO | ITD_BiO | MRBO | FSLA | VRBO | VR-saBiAdam | ASBiO-BreD |
|------|---------|---------|------|------|------|-------------|------------|
| 20s | 0.6509 | 0.6411 | 0.6103 | 0.6539 | 0.5951 | 0.6812 | 0.6653 |
| 40s | 0.7365 | 0.7210 | 0.6971 | 0.7399 | 0.6805 | 0.7141 | 0.7403 |
| 60s | 0.7762 | 0.7721 | 0.7519 | 0.7661 | 0.7429 | 0.7523 | **0.7830** |

Table 4: Validation accuracy *vs.* Running Time (5-way-5-shot) for different methods (with $L_1$ regularization)

| Time | AID_BiO | ITD_BiO | MRBO | FSLA | VRBO | VR-saBiAdam | ASBiO-BreD |
|------|---------|---------|------|------|------|-------------|------------|
| 20s | 0.8316 | 0.8131 | 0.8174 | 0.7993 | 0.7730 | 0.7753 | 0.8529 |
| 40s | 0.8779 | 0.8621 | 0.8634 | 0.8485 | 0.8305 | 0.8188 | 0.8967 |
| 60s | 0.9032 | 0.8968 | 0.8819 | 0.8824 | 0.8745 | 0.8640 | **0.9313** |

where each contains 5000 images. A portion of the training data are corrupted by randomly changing their labels, and we denote the portion of corrupted images as $\varrho$.

The detailed experimental setup is described in the Appendix A.1. For hyper-parameters, we perform grid search for our algorithms and other baselines to choose the best setting. The experimental results are summarized in Figure 1. As shown by the figure, BiO-BreD outperforms the reverse algorithm; SBiO-BreD outperforms AID-FP/stocBiO and AID-CG methods, and ASBiO-BreD outperforms the other SPIDER based algorithm MRBO and several momentum-based variance reduction methods: MRBO, SUSTAIN, FSLA, and VR-saBiAdam.

### 6.2 Hyper-representation Learning

In this subsection, we perform the hyper-representation learning task over the Omniglot dataset [24]. The formulation of this problem is as follows:

$$\min_\lambda \ l_{val}\big(\lambda, w^*(\lambda)\big) := \mathbb{E}_\xi \Big[ \frac{1}{|D_{\mathcal{V},\xi}|} \sum_{(x_i,y_i) \in D_{\mathcal{V},\xi}} l\Big( w_\xi^*(\lambda)^T \phi(x_i; \lambda), y_i\Big); \xi \Big] + \alpha \|\lambda\|_1$$

$$\text{s.t. } w_\xi^*(\lambda) = \arg\min_w \ l_{tr}(\lambda, w; \xi) := \frac{1}{|D_{\mathcal{T},\xi}|} \sum_{(x_i,y_i) \in D_{\mathcal{T},\xi}} l\Big( w^T \phi(x_i; \lambda), y_i\Big) + C\|w\|^2,$$

where $l(\cdot)$ denotes the cross entropy loss, $D_{\mathcal{T},\xi}$ and $D_{\mathcal{V},\xi}$ are training and validation datasets for randomly sampled meta task $\xi$. Here $\phi(\cdot, \cdot)$ is a four-layers convolutional neural network with max-pooling and 32 filters per layer [9], which denotes a representation mapping. $\lambda$ denotes the parameter vector of the representation mapping $\phi(\cdot, \cdot)$, and $C \geq 0$ is a tuning parameter to guarantee the inner problem to be strongly convex. The term $\alpha \|\lambda\|_1$ imposes the sparsity of hyper-representations. In the experiment, we set $\alpha = 0.001$ and $C = 0.01$.

The detailed experimental setup is described in the Appendix A.2. The results of validation accuracy (test accuracy) are summarized in Table 3 and 4. From these results, our ASBiO-BreD algorithm outperforms other baselines in the non-smooth case. We also consider the smooth case, where the upper level problem is not added the $L_1$ regularization. The results without $L_1$ regularization are given in the Appendix A.2.

## 7  Conclusions

In the paper, we proposed a class of enhanced bilevel optimization methods based on the Bregman distance to solve the nonconvex-strongly-convex bilevel optimization problems possibly with nonsmooth regularization. Moreover, we provided a comprehensive theoretical analysis framework to analyze our methods. The theoretical results show that our methods outperform the best known computational complexities with respect to the condition number $\kappa$ and the target accuracy $\epsilon$ for finding an $\epsilon$-stationary point.

## Acknowledgments and Disclosure of Funding

This work was partially supported by NSF IIS 1838627, 1837956, 1956002, 2211492, CNS 2213701, CCF 2217003, DBI 2225775. FH was a postdoctoral researcher at the University of Pittsburgh. FH and HH are the corresponding authors.

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
