# A  Experimental Details

In this section, we introduce more details of our experiments. we compare our algorithms (*i.e.*, BiO-BreD, SBiO-BreD, and ASBiO-BreD) with the following bilevel optimization algorithms: reverse [9]/AID-BiO [11, 22], AID-CG [12], AID-FP [12], stocBiO [22]), MRBO [21], VRBO [21], FSLA [28], SUSTAIN [23], and VR-saBiAdam [18]. We do not include results for STABLE [5]/SVRB [14], because they require matrix inversion which does not make sufficient progress compared to other baselines within a given time range. SMB/SEMA [13] method resembles SUSTAIN, thus we do not include it in the comparison.

## A.1  Data Hyper-cleaning

In this subsection, we perform data hyper-cleaning over the MNIST dataset [25]. The formulation of this problem is as follows:

$$\min_{\lambda} \, l_{val}\big(\lambda, w^*(\lambda)\big) := \frac{1}{|D_{\mathcal{V}}|} \sum_{(x_i,y_i)\in D_{\mathcal{V}}} l\big(x_i^T w^*(\lambda), y_i\big)$$

$$\text{s.t. } w^*(\lambda) = \arg\min_{w} \, l_{tr}(\lambda, w) := \frac{1}{|D_{\mathcal{T}}|} \sum_{(x_i,y_i)\in D_{\mathcal{T}}} \sigma(\lambda_i) l(x_i^T w, y_i) + C\|w\|^2,$$

where $l(\cdot)$ denotes the cross entropy loss, $D_{\mathcal{T}}$ and $D_{\mathcal{V}}$ are training and validation dataset, respectively. Here $\lambda = \{\lambda_i\}_{i\in\mathcal{D}_{\mathcal{T}}}$ are hyper-parameters and $C \geq 0$ is a tuning parameter, $\sigma(\cdot)$ denotes the sigmoid function. In experiment, we set $C = 0.001$.

For training/validation batch-size, we use batch-size of 32, while for VRBO and our ASBiO-BreD, we choose larger batch-size 5000 (parameter $b$ in Algorithm 3) and sampling interval (parameter $q$ in Algorithm 3) is set as 3. For stocBiO/AID-FP, AID-CG and reverse, we use the warm-start trick as our BiO-BreD algorithm, *i.e.* the inner variable starts from the state of last iteration (Line 4 of Algorithm 1). We fine tune the number of inner-loop iterations and set it to be 50 for these algorithms. For MRBO, VRBO, SUSTAIN and our SBiO-BreD/ASBiO-BreD, we set $K = 3$ to evaluate the hyper-gradient. For FSLA, $K = 1$ as the hyper-gradient is evaluated recursively. As for learning rates, we set 1000 as the outer learning rate for all algorithms except our algorithms which use 0.1 as we change the learning rate adaptively. As for the inner learning rates, we set the stepsize as 0.05 for reverse, BiO-BreD, AID-CG, stocBiO/AID-FP, MRBO/SUSTAIN, FSLA and our SBiO-BreD; we set the stepsize as 0.2 for VRBO, VR-saBiAdam and our ASBiO-BreD; we set the stepsize as 1 for SUSTAIN.

## A.2  Hyper-representation Learning

In this subsection, we perform the hyper-representation learning task over the Omniglot dataset [24]. The formulation of this problem (without $L_1$ regularization) is as follows:

$$\min_{\lambda} \, l_{val}\big(\lambda, w^*(\lambda)\big) := \mathbb{E}_{\xi}\Big[ \frac{1}{|D_{\mathcal{V},\xi}|} \sum_{(x_i,y_i)\in D_{\mathcal{V},\xi}} l\Big( w_{\xi}^*(\lambda)^T \phi(x_i; \lambda), y_i \Big); \xi \Big]$$

$$\text{s.t. } w_{\xi}^*(\lambda) = \arg\min_{w} \, l_{tr}(\lambda, w; \xi) := \frac{1}{|D_{\mathcal{T},\xi}|} \sum_{(x_i,y_i)\in D_{\mathcal{T},\xi}} l\Big( w^T \phi(x_i; \lambda), y_i \Big) + C\|w\|^2,$$

where $l(\cdot)$ denotes the cross entropy loss, $D_{\mathcal{T},\xi}$ and $D_{\mathcal{V},\xi}$ are training and validation dataset for randomly sampled meta task $\xi$. Here $\phi(\cdot, \cdot)$ is a four-layers convolutional neural network with maxpooling and 32 filters per layer [9], which denotes a representation mapping. $\lambda$ denotes the parameter vector of the representation mapping $\phi(\cdot, \cdot)$, and $C \geq 0$ is a tuning parameter to guarantee the inner problem to be strongly convex. In the experiment, we set $C = 0.01$.

In every hyper-iteration, we choose 4 meta tasks, while for VRBO and our ASBiO-BreD, we choose larger batch-size 16 (parameter $b$ in Algorithm 3) and sampling interval (parameter $q$ in Algorithm 3) is set as 3. For stocBiO/AID-FP, AID-CG and reverse, we use the warm-start trick as our BiO-BreD algorithm, *i.e.* the inner variable starts from the state of last iteration (Line 4 of Algorithm 1). We fine tune the number of inner-loop iterations and set it to be 16 for these algorithms. For MRBO, VRBO, SUSTAIN and our SBiO-BreD/ASBiO-BreD, we set $K = 5$ to evaluate the hyper-gradient. For FSLA, $K = 1$ as the hyper-gradient is evaluated recursively. As for learning rates, we set 1000 as the outer learning rate for all algorithms except our algorithms which use 0.001 as we change the learning rate adaptively. As for the inner learning rates, we set the stepsize as 0.4 for all algorithms.

The experimental results are summarized in Figure 2. As shown by the figure, BiO-BreD outperforms the reverse algorithm; SBiO-BreD outperforms AID-FP/stocBiO and AID-CG methods, while SBiO-BreD outperforms another SPIDER based algorithm MRBO and several momentum-based variance reduction methods: MRBO, SUSTAIN, FSLA and VR-saBiAdam.

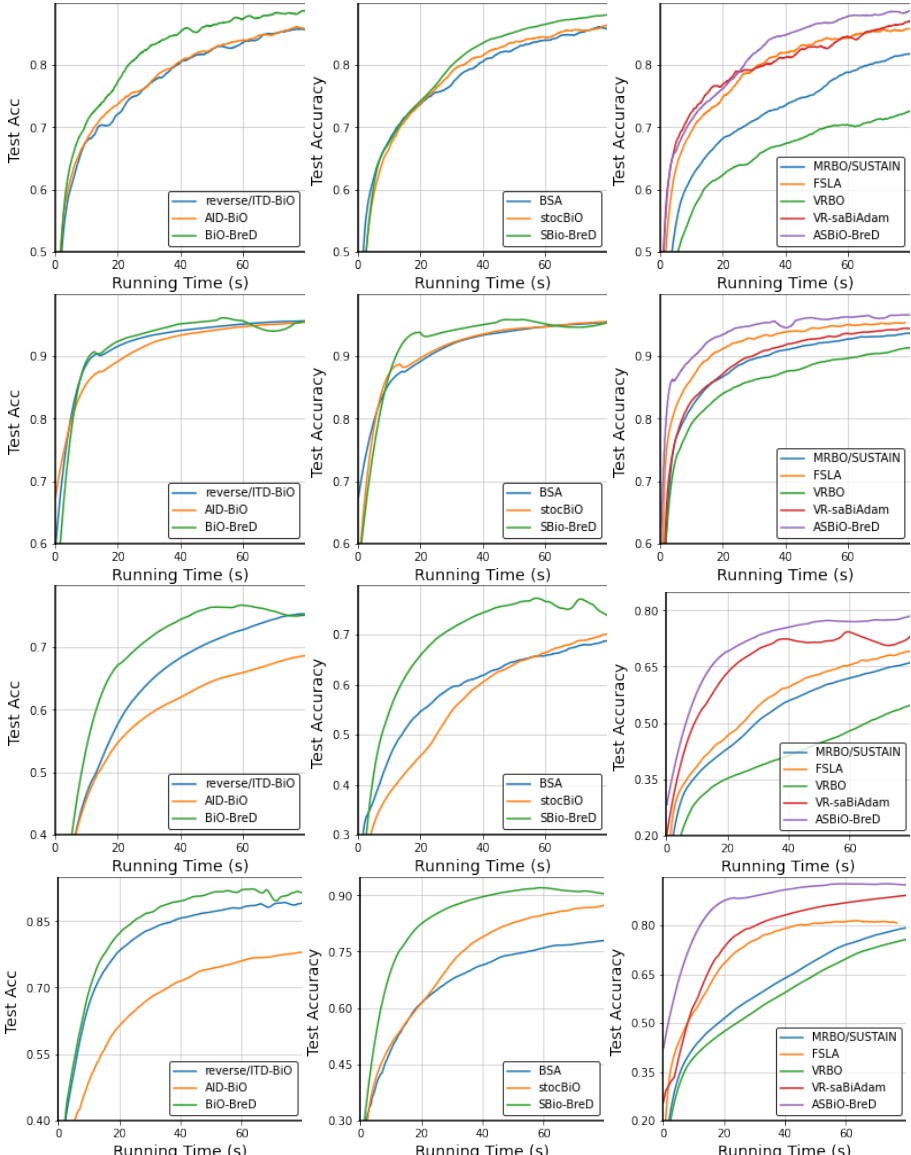

Figure 2: Validation Accuracy (Test Accuracy) $vs$. Running Time for different methods for the Omniglot Dataset. We compare our BiO-BreD with deterministic baselines (the first column), SBiO-BreD with stochastic baselines (the second column); ASBiO-BreD with momentum-based or SPIDER/SARAH based baselines (the last column). The first row shows results for 5-way-1-shot case; the second row shows results for 5-way-5-shot case; the third row shows results for 20-way-1-shot case; the last row shows results for 20-way-5-shot case.

# B    Detailed Convergence Analysis

In this section, we provide the detailed convergence analysis of our algorithms. We first gives some useful lemmas.

**Lemma 6.** *(Proposition 2. [22]) The gradient $\frac{\partial f(x_t, y_t^K)}{\partial x}$ is the following analytical form:*

$$\frac{\partial f(x_t, y_t^K)}{\partial x} = \nabla_x f(x_t, y_t^K) - \lambda \sum_{k=0}^{K-1} \nabla_{xy}^2 g(x_t, y_t^k) \prod_{j=k+1}^{K-1} \left( I_{d_2} - \lambda \nabla_{yy}^2 g(x_t, y_t^j) \right) \nabla_y f(x_t, y_t^K).$$

The above lemma 6 shows an analytical form of $w_t$ in Algorithm 1.

**Lemma 7.** *(Lemma 6. [22]) Under the above Assumptions, given the sequence $\{x_t, y_t\}_{t=1}^T$ generated from Algorithm 1, and $0 < \lambda < \frac{1}{L}$, we have*

$$\left\|\frac{\partial f(x_t, y_t^K)}{\partial x} - \nabla F(x_t)\right\| \leq \left(L_1(1-\lambda\mu)^{\frac{K}{2}} + L_2(1-\lambda\mu)^{\frac{K-1}{2}}\right)\|y_t^0 - y^*(x_t)\| + L_3(1-\lambda\mu)^K, \quad (10)$$

*where $L_1 = \frac{L(L+\mu)}{\mu}$, $L_2 = \frac{2C_{fy}(\mu L_{gxy} + LL_{gyy})}{\mu^2}$ and $L_3 = \frac{LC_{fy}}{\mu}$.*

The above lemma 7 shows the variance of gradient estimator $w_t = \frac{\partial f(x_t, y_t^K)}{\partial x}$ decays exponentially fast with iteration number $K$.

**Lemma 8.** *Given i.i.d. random variables $\{\zeta_i\}_{i=1}^n$ with zero mean, we have $\mathbb{E}\|\frac{1}{n}\sum_{i=1}^n \zeta_i\|^2 = \frac{1}{n}\mathbb{E}\|\zeta_i\|^2$ for any $i \in [n]$.*

**Lemma 9.** *(Lemma 1 in [10]) Let $x_{t+1} = \arg\min_{x\in\mathcal{X}}\left\{\langle w_t, x\rangle + h(x) + \frac{1}{\gamma}D_{\psi_t}(x, x_t)\right\}$ and $\tilde{\mathcal{G}}_t = \frac{1}{\gamma}(x_t - x_{t+1})$, we have, for all $t \geq 1$*

$$\langle w_t, \tilde{\mathcal{G}}_t\rangle \geq \rho\|\tilde{\mathcal{G}}_t\|^2 + \frac{1}{\gamma}\left(h(x_{t+1}) - h(x_t)\right), \quad (11)$$

*where $\rho > 0$ depends on $\rho$-strongly convex function $\psi_t(x)$.*

**Lemma 10.** *(Lemma 2 in [10]) Let $\{x_t\}_{t=1}^T$ be generated from Algorithms 1, 2 and 3, and define $x_{t+1}^+ = \arg\min_{x\in\mathcal{X}}\left\{\langle \nabla F(x_t), x\rangle + h(x) + \frac{1}{\gamma}D_{\psi_t}(x, x_t)\right\}$, and let $\mathcal{G}_t = \frac{1}{\gamma}(x_t - x_{t+1}^+)$, $\tilde{\mathcal{G}}_t = \frac{1}{\gamma}(x_t - x_{t+1})$, we have*

$$\|\mathcal{G}_t - \tilde{\mathcal{G}}_t\| \leq \frac{1}{\rho}\|\nabla F(x_t) - w_t\|, \quad (12)$$

*where $F(x_t) = f(x_t, y^*(x_t))$ and $\rho > 0$ depends on $\rho$-strongly convex function $\psi_t(x)$.*

**Lemma 11.** *(Restatement of Lemma 5) Suppose the sequence $\{x_t, y_t\}_{t=1}^T$ be generated from Algorithms 2 and 3. Under the above assumptions, given $0 < \eta_t \leq 1$ for all $t \geq 1$ and $0 < \lambda \leq \frac{1}{6L}$, we have*

$$\|y_{t+1} - y^*(x_{t+1})\|^2 \leq (1 - \frac{\eta_t\mu\lambda}{4})\|y_t - y^*(x_t)\|^2 - \frac{3\eta_t\lambda^2}{4}\|v_t\|^2$$
$$+ \frac{25\eta_t\lambda}{6\mu}\|\nabla_y g(x_t, y_t) - v_t\|^2 + \frac{25\kappa^2}{6\eta_t\mu\lambda}\|x_{t+1} - x_t\|^2, \quad (13)$$

*where $\kappa = L/\mu$.*

*Proof.* We first use the step $y_{t+1} = y_t + \eta_t(\tilde{y}_{t+1} - y_t)$ and $\tilde{y}_{t+1} = y_t - \lambda v_t$ instead of the step 5 in Algorithm 2 and step 11 in Algorithm 3, i.e., $y_{t+1} = y_t - \lambda\eta_t v_t$. This proof mainly follows the proof of Lemma 28 in [17].

According to Assumption 1, i.e., the function $g(x, y)$ is $\mu$-strongly convex w.r.t $y$, we have

$$g(x_t, y) \geq g(x_t, y_t) + \langle\nabla_y g(x_t, y_t), y - y_t\rangle + \frac{\mu}{2}\|y - y_t\|^2$$
$$= g(x_t, y_t) + \langle v_t, y - \tilde{y}_{t+1}\rangle + \langle\nabla_y g(x_t, y_t) - v_t, y - \tilde{y}_{t+1}\rangle$$
$$+ \langle\nabla_y g(x_t, y_t), \tilde{y}_{t+1} - y_t\rangle + \frac{\mu}{2}\|y - y_t\|^2. \quad (14)$$

According to Assumption 2, i.e., the function $g(x, y)$ is $L$-smooth, we have

$$g(x_t, \tilde{y}_{t+1}) \leq g(x_t, y_t) + \langle\nabla_y g(x_t, y_t), \tilde{y}_{t+1} - y_t\rangle + \frac{L}{2}\|\tilde{y}_{t+1} - y_t\|^2. \quad (15)$$

By combining the about inequalities (14) with (15), we have

$$g(x_t, y) \geq g(x_t, \tilde{y}_{t+1}) + \langle v_t, y - \tilde{y}_{t+1}\rangle + \langle\nabla_y g(x_t, y_t) - v_t, y - \tilde{y}_{t+1}\rangle$$
$$+ \frac{\mu}{2}\|y - y_t\|^2 - \frac{L}{2}\|\tilde{y}_{t+1} - y_t\|^2. \quad (16)$$

According to $\tilde{y}_{t+1} = y_t - \lambda v_t$, we have

$$\langle v_t, y - \tilde{y}_{t+1}\rangle = \frac{1}{\lambda}\langle\tilde{y}_{t+1} - y_t, \tilde{y}_{t+1} - y\rangle$$
$$= \frac{1}{\lambda}\|\tilde{y}_{t+1} - y_t\|^2 + \frac{1}{\lambda}\langle\tilde{y}_{t+1} - y_t, y_t - y\rangle. \quad (17)$$

By plugging the inequalities (17) into (16), we have

$$g(x_t, y) \geq g(x_t, \tilde{y}_{t+1}) + \frac{1}{\lambda}\langle \tilde{y}_{t+1} - y_t, y_t - y \rangle + \frac{1}{\lambda}\|\tilde{y}_{t+1} - y_t\|^2$$
$$+ \langle \nabla_y g(x_t, y_t) - v_t, y - \tilde{y}_{t+1} \rangle + \frac{\mu}{2}\|y - y_t\|^2 - \frac{L}{2}\|\tilde{y}_{t+1} - y_t\|^2. \tag{18}$$

Let $y = y^*(x_t)$, then we have

$$g(x_t, y^*(x_t)) \geq g(x_t, \tilde{y}_{t+1}) + \frac{1}{\lambda}\langle \tilde{y}_{t+1} - y_t, y_t - y^*(x_t) \rangle + (\frac{1}{\lambda} - \frac{L}{2})\|\tilde{y}_{t+1} - y_t\|^2$$
$$+ \langle \nabla_y g(x_t, y_t) - v_t, y^*(x_t) - \tilde{y}_{t+1} \rangle + \frac{\mu}{2}\|y^*(x_t) - y_t\|^2. \tag{19}$$

Due to the strongly-convexity of $g(\cdot, y)$ and $y^*(x_t) = \arg\min_{y \in \mathcal{Y}} g(x_t, y)$, we have $g(x_t, y^*(x_t)) \leq g(x_t, \tilde{y}_{t+1})$. Thus, we obtain

$$0 \geq \frac{1}{\lambda}\langle \tilde{y}_{t+1} - y_t, y_t - y^*(x_t) \rangle + \langle \nabla_y g(x_t, y_t) - v_t, y^*(x_t) - \tilde{y}_{t+1} \rangle$$
$$+ (\frac{1}{\lambda} - \frac{L}{2})\|\tilde{y}_{t+1} - y_t\|^2 + \frac{\mu}{2}\|y^*(x_t) - y_t\|^2. \tag{20}$$

By $y_{t+1} = y_t + \eta_t(\tilde{y}_{t+1} - y_t)$, we have

$$\|y_{t+1} - y^*(x_t)\|^2 = \|y_t + \eta_t(\tilde{y}_{t+1} - y_t) - y^*(x_t)\|^2$$
$$= \|y_t - y^*(x_t)\|^2 + 2\eta_t\langle \tilde{y}_{t+1} - y_t, y_t - y^*(x_t) \rangle + \eta_t^2\|\tilde{y}_{t+1} - y_t\|^2. \tag{21}$$

Then we obtain

$$\langle \tilde{y}_{t+1} - y_t, y_t - y^*(x_t) \rangle = \frac{1}{2\eta_t}\|y_{t+1} - y^*(x_t)\|^2 - \frac{1}{2\eta_t}\|y_t - y^*(x_t)\|^2 - \frac{\eta_t}{2}\|\tilde{y}_{t+1} - y_t\|^2. \tag{22}$$

Consider the upper bound of the term $\langle \nabla_y g(x_t, y_t) - v_t, y^*(x_t) - \tilde{y}_{t+1} \rangle$, we have

$$\langle \nabla_y g(x_t, y_t) - v_t, y^*(x_t) - \tilde{y}_{t+1} \rangle$$
$$= \langle \nabla_y g(x_t, y_t) - v_t, y^*(x_t) - y_t \rangle + \langle \nabla_y g(x_t, y_t) - v_t, y_t - \tilde{y}_{t+1} \rangle$$
$$\geq -\frac{1}{\mu}\|\nabla_y g(x_t, y_t) - v_t\|^2 - \frac{\mu}{4}\|y^*(x_t) - y_t\|^2 - \frac{1}{\mu}\|\nabla_y g(x_t, y_t) - v_t\|^2 - \frac{\mu}{4}\|y_t - \tilde{y}_{t+1}\|^2$$
$$= -\frac{2}{\mu}\|\nabla_y g(x_t, y_t) - v_t\|^2 - \frac{\mu}{4}\|y^*(x_t) - y_t\|^2 - \frac{\mu}{4}\|y_t - \tilde{y}_{t+1}\|^2. \tag{23}$$

By plugging the inequalities (22) and (23) into (20), we obtain

$$\frac{1}{2\eta_t\lambda}\|y_{t+1} - y^*(x_t)\|^2$$
$$\leq (\frac{1}{2\eta_t\lambda} - \frac{\mu}{4})\|y_t - y^*(x_t)\|^2 + (\frac{\eta_t}{2\lambda} + \frac{\mu}{4} + \frac{L}{2} - \frac{1}{\lambda})\|\tilde{y}_{t+1} - y_t\|^2 + \frac{2}{\mu}\|\nabla_y g(x_t, y_t) - v_t\|^2$$
$$\leq (\frac{1}{2\eta_t\lambda} - \frac{\mu}{4})\|y_t - y^*(x_t)\|^2 + (\frac{3L}{4} - \frac{1}{2\lambda})\|\tilde{y}_{t+1} - y_t\|^2 + \frac{2}{\mu}\|\nabla_y g(x_t, y_t) - v_t\|^2$$
$$= (\frac{1}{2\eta_t\lambda} - \frac{\mu}{4})\|y_t - y^*(x_t)\|^2 - (\frac{3}{8\lambda} + \frac{1}{8\lambda} - \frac{3L}{4})\|\tilde{y}_{t+1} - y_t\|^2 + \frac{2}{\mu}\|\nabla_y g(x_t, y_t) - v_t\|^2$$
$$\leq (\frac{1}{2\eta_t\lambda} - \frac{\mu}{4})\|y_t - y^*(x_t)\|^2 - \frac{3}{8\lambda}\|\tilde{y}_{t+1} - y_t\|^2 + \frac{2}{\mu}\|\nabla_y g(x_t, y_t) - v_t\|^2, \tag{24}$$

where the second inequality holds by $L \geq \mu$ and $0 < \eta_t \leq 1$, and the last inequality is due to $0 < \lambda \leq \frac{1}{6L}$. It implies that

$$\|y_{t+1} - y^*(x_t)\|^2 \leq (1 - \frac{\eta_t\mu\lambda}{2})\|y_t - y^*(x_t)\|^2 - \frac{3\eta_t}{4}\|\tilde{y}_{t+1} - y_t\|^2 + \frac{4\eta_t\lambda}{\mu}\|\nabla_y g(x_t, y_t) - v_t\|^2. \tag{25}$$

Next, we decompose the term $\|y_{t+1} - y^*(x_{t+1})\|^2$ as follows:

$$\|y_{t+1} - y^*(x_{t+1})\|^2 = \|y_{t+1} - y^*(x_t) + y^*(x_t) - y^*(x_{t+1})\|^2$$
$$= \|y_{t+1} - y^*(x_t)\|^2 + 2\langle y_{t+1} - y^*(x_t), y^*(x_t) - y^*(x_{t+1}) \rangle + \|y^*(x_t) - y^*(x_{t+1})\|^2$$
$$\leq (1 + \frac{\eta_t\mu\lambda}{4})\|y_{t+1} - y^*(x_t)\|^2 + (1 + \frac{4}{\eta_t\mu\lambda})\|y^*(x_t) - y^*(x_{t+1})\|^2$$
$$\leq (1 + \frac{\eta_t\mu\lambda}{4})\|y_{t+1} - y^*(x_t)\|^2 + (1 + \frac{4}{\eta_t\mu\lambda})\kappa^2\|x_t - x_{t+1}\|^2, \tag{26}$$

where the first inequality holds by Cauchy-Schwarz inequality and Young's inequality, and the second inequality is due to Lemma 2, and the last equality holds by $x_{t+1} = x_t + \eta_t(\tilde{x}_{t+1} - x_t)$.

By combining the above inequalities (25) and (26), we have

$$\|y_{t+1} - y^*(x_{t+1})\|^2 \leq (1 + \frac{\eta_t\mu\lambda}{4})(1 - \frac{\eta_t\mu\lambda}{2})\|y_t - y^*(x_t)\|^2 - (1 + \frac{\eta_t\mu\lambda}{4})\frac{3\eta_t}{4}\|\tilde{y}_{t+1} - y_t\|^2$$
$$+ (1 + \frac{\eta_t\mu\lambda}{4})\frac{4\eta_t\lambda}{\mu}\|\nabla_y g(x_t, y_t) - v_t\|^2 + (1 + \frac{4}{\eta_t\mu\lambda})\kappa^2\|x_t - x_{t+1}\|^2.$$

Since $0 < \eta_t \leq 1$, $0 < \lambda \leq \frac{1}{6L}$ and $L \geq \mu$, we have $\lambda \leq \frac{1}{6L} \leq \frac{1}{6\mu}$ and $\eta_t \leq 1 \leq \frac{1}{6\mu\lambda}$. Then by using $\geq 1$, we have

$$(1 + \frac{\eta_t\mu\lambda}{4})(1 - \frac{\eta_t\mu\lambda}{2}) = 1 - \frac{\eta_t\mu\lambda}{2} + \frac{\eta_t\mu\lambda}{4} - \frac{\eta_t^2\mu^2\lambda^2}{8} \leq 1 - \frac{\eta_t\mu\lambda}{4},$$
$$-(1 + \frac{\eta_t\mu\lambda}{4})\frac{3\eta_t}{4} \leq -\frac{3\eta_t}{4},$$
$$(1 + \frac{\eta_t\mu\lambda}{4})\frac{4\eta_t\lambda}{\mu} \leq (1 + \frac{1}{24})\frac{4\eta_t\lambda}{\mu} = \frac{25\eta_t\lambda}{6\mu},$$
$$(1 + \frac{4}{\eta_t\mu\lambda})\kappa^2 \leq \kappa^2 + \frac{4\kappa^2}{\eta_t\mu\lambda} \leq \frac{\kappa^2}{6\eta_t\mu\lambda} + \frac{4\kappa^2}{\eta_t\mu\lambda} = \frac{25\kappa^2}{6\eta_t\mu\lambda}.$$

Thus we have

$$\|y_{t+1} - y^*(x_{t+1})\|^2 \leq (1 - \frac{\eta_t\mu\lambda}{4})\|y_t - y^*(x_t)\|^2 - \frac{3\eta_t\lambda^2}{4}\|v_t\|^2$$
$$+ \frac{25\eta_t\lambda}{6\mu}\|\nabla_y g(x_t, y_t) - v_t\|^2 + \frac{25\kappa^2}{6\eta_t\mu\lambda}\|x_{t+1} - x_t\|^2. \tag{27}$$

$\square$

## B.1 Convergence Analysis of the BiO-BreD Algorithm

In this subsection, we provide the convergence analysis of our BiO-BreD algorithm.

**Theorem 4.** *(Restatement of Theorem 1) Suppose the sequence* $\{x_t, y_t\}_{t=1}^T$ *be generated from Algorithm 1. Let* $0 < \gamma \leq \frac{3\rho}{4L_F}$, $0 < \lambda < \frac{1}{L}$, $K = \log(T)/\log(\frac{1}{1-\lambda\mu}) + 1$ *and* $\|y_t^0 - y^*(x_t)\|^2 \leq \Delta$ *for all* $t \geq 0$, *we have*

$$\frac{1}{T}\sum_{t=0}^{T-1}\|\mathcal{G}_t\|^2 \leq \frac{16(\Phi(x_0) - \Phi^*)}{3T\gamma\rho} + \frac{22\Delta L_1^2}{\rho^2 T} + \frac{22\Delta L_2^2}{\rho^2 T} + \frac{22L_3^2}{\rho^2 T^2}, \tag{28}$$

*where* $L_1 = \frac{L(L+\mu)}{\mu}$, $L_2 = \frac{2C_{fy}(\mu L_{gxy} + LL_{gyy})}{\mu^2}$ *and* $L_3 = \frac{LC_{fy}}{\mu}$.

*Proof.* According to the above Lemma 2, the function $F(x)$ has $L_F$-Lipschitz continuous gradient. Let $\tilde{\mathcal{G}}_t = \frac{1}{\gamma}(x_t - x_{t+1})$, we have

$$F(x_{t+1}) \leq F(x_t) + \langle \nabla F(x_t), x_{t+1} - x_t \rangle + \frac{L_F}{2}\|x_{t+1} - x_t\|^2$$
$$= F(x_t) - \gamma\langle \nabla F(x_t), \tilde{\mathcal{G}}_t \rangle + \frac{\gamma^2 L_F}{2}\|\tilde{\mathcal{G}}_t\|^2$$
$$= F(x_t) - \gamma\langle \frac{\partial f(x_t, y_t^K)}{\partial x}, \tilde{\mathcal{G}}_t \rangle + \gamma\langle \frac{\partial f(x_t, y_t^K)}{\partial x} - \nabla F(x_t), \tilde{\mathcal{G}}_t \rangle + \frac{\gamma^2 L_F}{2}\|\tilde{\mathcal{G}}_t\|^2$$
$$\leq F(x_t) - \gamma\rho\|\tilde{\mathcal{G}}_t\|^2 - h(x_{t+1}) + h(x_t) + \gamma\langle \frac{\partial f(x_t, y_t^K)}{\partial x} - \nabla F(x_t), \tilde{\mathcal{G}}_t \rangle + \frac{\gamma^2 L_F}{2}\|\tilde{\mathcal{G}}_t\|^2$$
$$\leq F(x_t) + (\frac{\gamma^2 L_F}{2} - \frac{3\gamma\rho}{4})\|\tilde{\mathcal{G}}_t\|^2 - h(x_{t+1}) + h(x_t) + \frac{\gamma}{\rho}\|\frac{\partial f(x_t, y_t^K)}{\partial x} - \nabla F(x_t)\|^2, \tag{29}$$

where the second last inequality holds by the above Lemma 9, and the last inequality holds by the following inequality

$$\langle \frac{\partial f(x_t, y_t^K)}{\partial x} - \nabla F(x_t), \tilde{\mathcal{G}}_t \rangle \leq \|\frac{\partial f(x_t, y_t^K)}{\partial x} - \nabla F(x_t)\|\|\tilde{\mathcal{G}}_t\|$$
$$\leq \frac{1}{\rho}\|\frac{\partial f(x_t, y_t^K)}{\partial x} - \nabla F(x_t)\|^2 + \frac{\rho}{4}\|\tilde{\mathcal{G}}_t\|^2. \tag{30}$$

According to the above Lemma 7, we have

$$\|\frac{\partial f(x_t, y_t^K)}{\partial x} - \nabla F(x_t)\|^2$$
$$\leq 3\big(L_1^2(1-\lambda\mu)^K + L_2^2(1-\lambda\mu)^{K-1}\big)\|y_t^0 - y^*(x_t)\|^2 + 3L_3^2(1-\lambda\mu)^{2K}$$
$$\leq 3\Delta L_1^2(1-\lambda\mu)^K + 3\Delta L_2^2(1-\lambda\mu)^{K-1} + 3L_3^2(1-\lambda\mu)^{2K}, \tag{31}$$

where the last inequality holds by $\|y_t^0 - y^*(x_t)\|^2 \leq \Delta$ for all $t \geq 0$.

Let $\Phi(x) = F(x) + h(x)$, plugging (31) into (29), we have

$$\Phi(x_{t+1}) \leq \Phi(x_t) + (\frac{\gamma^2 L_F}{2} - \frac{3\gamma\rho}{4})\|\tilde{\mathcal{G}}_t\|^2 + \frac{\gamma}{\rho}\|\frac{\partial f(x_t, y_t^K)}{\partial x} - \nabla F(x_t)\|^2$$
$$\leq \Phi(x_t) - \frac{3\gamma\rho}{8}\|\tilde{\mathcal{G}}_t\|^2 + \frac{3\gamma\Delta L_1^2}{\rho}(1-\lambda\mu)^K + \frac{3\gamma\Delta L_2^2}{\rho}(1-\lambda\mu)^{K-1} + \frac{3\gamma L_3^2}{\rho}(1-\lambda\mu)^{2K}, \tag{32}$$

where the last inequality is due to $0 < \gamma \leq \frac{3\rho}{4L_F}$ and the above inequality (31). According to Lemma 10, the difference between $\tilde{\mathcal{G}}_t$ and $\mathcal{G}_t$ are bounded, we have

$$\|\mathcal{G}_t\|^2 \leq 2\|\tilde{\mathcal{G}}_t\|^2 + 2\|\tilde{\mathcal{G}}_t - \mathcal{G}_t\|^2$$
$$\leq 2\|\tilde{\mathcal{G}}_t\|^2 + \frac{2}{\rho^2}\|w_t - \nabla F(x_t)\|^2$$
$$= 2\|\tilde{\mathcal{G}}_t\|^2 + \frac{2}{\rho^2}\|\frac{\partial f(x_t, y_t^K)}{\partial x} - \nabla F(x_t)\|^2$$
$$\leq 2\|\tilde{\mathcal{G}}_t\|^2 + \frac{6\Delta L_1^2}{\rho^2}(1-\lambda\mu)^K + \frac{6\Delta L_2^2}{\rho^2}(1-\lambda\mu)^{K-1} + \frac{6L_3^2}{\rho^2}(1-\lambda\mu)^{2K}. \tag{33}$$

Thus we have

$$-\|\tilde{\mathcal{G}}_t\|^2 \leq -\frac{1}{2}\|\mathcal{G}_t\|^2 + \frac{3\Delta L_1^2}{\rho^2}(1-\lambda\mu)^K + \frac{3\Delta L_2^2}{\rho^2}(1-\lambda\mu)^{K-1} + \frac{3L_3^2}{\rho^2}(1-\lambda\mu)^{2K}. \tag{34}$$

By plugging (34) into (29), we have

$$\Phi(x_{t+1}) \leq \Phi(x_t) - \frac{3\gamma\rho}{16}\|\mathcal{G}_t\|^2 + \frac{3\gamma\rho}{8}\big(\frac{3\Delta L_1^2}{\rho^2}(1-\lambda\mu)^K + \frac{3\Delta L_2^2}{\rho^2}(1-\lambda\mu)^{K-1} + \frac{3L_3^2}{\rho^2}(1-\lambda\mu)^{2K}\big)$$
$$+ \frac{3\gamma\Delta L_1^2}{\rho}(1-\lambda\mu)^K + \frac{3\gamma\Delta L_2^2}{\rho}(1-\lambda\mu)^{K-1} + \frac{3\gamma L_3^2}{\rho}(1-\lambda\mu)^{2K}$$
$$= \Phi(x_t) - \frac{3\gamma\rho}{16}\|\mathcal{G}_t\|^2 + \frac{33\gamma\Delta L_1^2}{8\rho}(1-\lambda\mu)^K + \frac{33\gamma\Delta L_2^2}{8\rho}(1-\lambda\mu)^{K-1} + \frac{33\gamma L_3^2}{8\rho}(1-\lambda\mu)^{2K}. \tag{35}$$

Thus, we have

$$\frac{1}{T}\sum_{t=0}^{T-1}\|\mathcal{G}_t\|^2 \leq \frac{16\big(\Phi(x_0) - \Phi(x_T)\big)}{3T\gamma\rho} + \frac{22\Delta L_1^2}{\rho^2}(1-\lambda\mu)^K + \frac{22\Delta L_2^2}{\rho^2}(1-\lambda\mu)^{K-1} + \frac{22L_3^2}{\rho^2}(1-\lambda\mu)^{2K}$$
$$\leq \frac{16\big(\Phi(x_0) - \Phi^*\big)}{3T\gamma\rho} + \frac{22\Delta L_1^2}{\rho^2}(1-\lambda\mu)^K + \frac{22\Delta L_2^2}{\rho^2}(1-\lambda\mu)^{K-1} + \frac{22L_3^2}{\rho^2}(1-\lambda\mu)^{2K}$$
$$\leq \frac{16\big(\Phi(x_0) - \Phi^*\big)}{3T\gamma\rho} + \frac{22\Delta L_1^2}{\rho^2 T} + \frac{22\Delta L_2^2}{\rho^2 T} + \frac{22L_3^2}{\rho^2 T^2}, \tag{36}$$

where the last inequality holds by $K = \log(T)/\log(\frac{1}{1-\lambda\mu}) + 1$ and $\lambda < \frac{1}{L}$.

$\square$

## B.2 Convergence Analysis of the SBiO-BreD Algorithm

In this subsection, we provide the convergence analysis of our SBiO-BreD algorithm. Let $R(x_t, y_t) = \bar{\nabla} f(x_t, y_t) - \bar{\nabla} f(x_t, y_t; \mathcal{B}_t)$ for all $t \geq 0$.

**Theorem 5.** *(Restatement of Theorem 2) Suppose the sequence $\{x_t, y_t\}_{t=1}^T$ be generated from Algorithm 2. Let $K = \frac{L}{\mu}\log(\frac{LC_{f_y}T}{\mu})$, $0 < \eta = \eta_t \leq 1$, $0 < \gamma \leq \min\big(\frac{3\rho}{4L_F}, \frac{9\eta\rho\mu\lambda}{800\kappa^2}, \frac{\eta\mu\rho\lambda}{47L_y^2}\big)$ and $0 < \lambda \leq \frac{1}{6L}$, we have*

$$\frac{1}{T}\sum_{t=1}^T \mathbb{E}\|\mathcal{G}_t\|^2 \leq \frac{32(\Phi(x_0) - \Phi^*)}{3T\gamma\rho} + \frac{32\Delta}{3T\gamma\rho} + \frac{752\sigma^2}{3\rho^2 b} + \frac{400\eta\lambda\sigma^2}{9\gamma\rho\mu b} + \frac{752}{3\rho^2 T^2}, \tag{37}$$

where $\Delta = \|y_0 - y^*(x_0)\|^2$.

*Proof.* According to the above Lemma 2, the function $F(x)$ has $L_F$-Lipschitz continuous gradient. Let $\tilde{\mathcal{G}}_t = \frac{1}{\gamma}(x_t - x_{t+1})$, we have

$$
\begin{aligned}
F(x_{t+1}) &\leq F(x_t) + \langle \nabla F(x_t), x_{t+1} - x_t \rangle + \frac{L_F}{2}\|x_{t+1} - x_t\|^2 \\
&= F(x_t) - \gamma\langle \nabla F(x_t), \tilde{\mathcal{G}}_t \rangle + \frac{\gamma^2 L_F}{2}\|\tilde{\mathcal{G}}_t\|^2 \\
&= F(x_t) - \gamma\langle w_t, \tilde{\mathcal{G}}_t \rangle + \gamma\langle w_t - \nabla F(x_t), \tilde{\mathcal{G}}_t \rangle + \frac{\gamma^2 L_F}{2}\|\tilde{\mathcal{G}}_t\|^2 \\
&\leq F(x_t) - \gamma\rho\|\tilde{\mathcal{G}}_t\|^2 - h(x_{t+1}) + h(x_t) + \gamma\langle w_t - \nabla F(x_t), \tilde{\mathcal{G}}_t \rangle + \frac{\gamma^2 L_F}{2}\|\tilde{\mathcal{G}}_t\|^2 \\
&\leq F(x_t) + \left(\frac{\gamma^2 L_F}{2} - \frac{3\gamma\rho}{4}\right)\|\tilde{\mathcal{G}}_t\|^2 - h(x_{t+1}) + h(x_t) + \frac{\gamma}{\rho}\|w_t - \nabla F(x_t)\|^2,
\end{aligned}
\tag{38}
$$

where the second last inequality holds by the above Lemma 9, and the last inequality holds by the following inequality

$$
\begin{aligned}
\langle w_t - \nabla F(x_t), \tilde{\mathcal{G}}_t \rangle &\leq \|w_t - \nabla F(x_t)\|\|\tilde{\mathcal{G}}_t\| \\
&\leq \frac{1}{\rho}\|w_t - \nabla F(x_t)\|^2 + \frac{\rho}{4}\|\tilde{\mathcal{G}}_t\|^2.
\end{aligned}
\tag{39}
$$

According to the above Lemma 2, we have

$$
\begin{aligned}
\|w_t - \nabla F(x_t)\|^2 &= \|w_t - \bar{\nabla}f(x_t, y_t) + \bar{\nabla}f(x_t, y_t) - \nabla F(x_t)\|^2 \\
&\leq 2\|w_t - \bar{\nabla}f(x_t, y_t)\|^2 + 2\|\bar{\nabla}f(x_t, y_t) - \nabla F(x_t)\|^2 \\
&\leq 2\|w_t - \bar{\nabla}f(x_t, y_t)\|^2 + 2L_y^2\|y_t - y^*(x_t)\|^2.
\end{aligned}
\tag{40}
$$

Let $\Phi(x) = F(x) + h(x)$, plugging (40) into (38), we have

$$
\begin{aligned}
\Phi(x_{t+1}) &\leq \Phi(x_t) + \left(\frac{\gamma^2 L_F}{2} - \frac{3\gamma\rho}{4}\right)\|\tilde{\mathcal{G}}_t\|^2 + \frac{2\gamma}{\rho}\|w_t - \nabla_x f(x_t, y_t)\|^2 + \frac{2L_y^2\gamma}{\rho}\|y_t - y^*(x_t)\|^2 \\
&\leq \Phi(x_t) - \frac{3\gamma\rho}{8}\|\tilde{\mathcal{G}}_t\|^2 + \frac{2\gamma}{\rho}\|w_t - \nabla_x f(x_t, y_t)\|^2 + \frac{2L_y^2\gamma}{\rho}\|y_t - y^*(x_t)\|^2,
\end{aligned}
\tag{41}
$$

where the last inequality is due to $0 < \gamma \leq \frac{3\rho}{4L_F}$. According to Lemma 10, the difference between $\tilde{\mathcal{G}}_t$ and $\mathcal{G}_t$ are bounded, we have

$$
\begin{aligned}
\|\mathcal{G}_t\|^2 &\leq 2\|\tilde{\mathcal{G}}_t\|^2 + 2\|\tilde{\mathcal{G}}_t - \mathcal{G}_t\|^2 \\
&\leq 2\|\tilde{\mathcal{G}}_t\|^2 + \frac{2}{\rho^2}\|w_t - \nabla F(x_t)\|^2 \\
&\leq 2\|\tilde{\mathcal{G}}_t\|^2 + \frac{4}{\rho^2}\|w_t - \bar{\nabla}f(x_t, y_t)\|^2 + \frac{4L_y^2}{\rho^2}\|y_t - y^*(x_t)\|^2.
\end{aligned}
\tag{42}
$$

Thus we have

$$
-\|\tilde{\mathcal{G}}_t\|^2 \leq -\frac{1}{2}\|\mathcal{G}_t\|^2 + \frac{2}{\rho^2}\|w_t - \bar{\nabla}f(x_t, y_t)\|^2 + \frac{2L_y^2}{\rho^2}\|y_t - y^*(x_t)\|^2.
\tag{43}
$$

By plugging (43) into (41), we have

$$
\begin{aligned}
\Phi(x_{t+1}) &\leq \Phi(x_t) - \frac{3\gamma\rho}{16}\|\mathcal{G}_t\|^2 + \frac{3\gamma\rho}{8}\left(\frac{2}{\rho^2}\|w_t - \bar{\nabla}f(x_t, y_t)\|^2 + \frac{2L_y^2}{\rho^2}\|y_t - y^*(x_t)\|^2\right) \\
&\quad + \frac{2\gamma}{\rho}\|w_t - \bar{\nabla}f(x_t, y_t)\|^2 + \frac{2L_y^2\gamma}{\rho}\|y_t - y^*(x_t)\|^2 \\
&= \Phi(x_t) - \frac{3\gamma\rho}{16}\|\mathcal{G}_t\|^2 + \frac{11\gamma}{4\rho}\|w_t - \bar{\nabla}f(x_t, y_t)\|^2 + \frac{11L_y^2\gamma}{4\rho}\|y_t - y^*(x_t)\|^2.
\end{aligned}
\tag{44}
$$

Next, we define a useful Lyapunov function, for any $t \geq 1$

$$
\Omega_t = \mathbb{E}\left[\Phi(x_t) + \|y_t - y^*(x_t)\|^2\right].
\tag{45}
$$

According to Lemma 11, we have

$$\|y_{t+1} - y^*(x_{t+1})\|^2 - \|y_t - y^*(x_t)\|^2 \leq -\frac{\eta_t \mu \lambda}{4}\|y_t - y^*(x_t)\|^2 - \frac{3\eta_t \lambda^2}{4}\|v_t\|^2$$
$$+ \frac{25\eta_t \lambda}{6\mu}\|\nabla_y g(x_t, y_t) - v_t\|^2 + \frac{25\kappa^2}{6\eta_t \mu \lambda}\|x_{t+1} - x_t\|^2. \quad (46)$$

Then we have

$$\Omega_{t+1} - \Omega_t = \mathbb{E}\big[\Phi(x_{t+1}) - \Phi(x_t) + \|y_{t+1} - y^*(x_{t+1})\|^2 - \|y_t - y^*(x_t)\|^2\big]$$

$$\leq -\frac{3\gamma\rho}{16}\mathbb{E}\|\mathcal{G}_t\|^2 + \frac{11\gamma}{4\rho}\mathbb{E}\|w_t - \bar{\nabla}f(x_t, y_t)\|^2 + \frac{11 L_y^2 \gamma}{4\rho}\mathbb{E}\|y_t - y^*(x_t)\|^2 - \frac{\eta_t \mu \lambda}{4}\mathbb{E}\|y_t - y^*(x_t)\|^2$$

$$- \frac{3\eta_t \lambda^2}{4}\mathbb{E}\|v_t\|^2 + \frac{25\eta_t \lambda}{6\mu}\mathbb{E}\|\nabla_y g(x_t, y_t) - v_t\|^2 + \frac{25\kappa^2}{6\eta_t \mu \lambda}\mathbb{E}\|x_{t+1} - x_t\|^2$$

$$= -\frac{3\gamma\rho}{16}\mathbb{E}\|\mathcal{G}_t\|^2 + \frac{11\gamma}{4\rho}\mathbb{E}\|w_t - \bar{\nabla}f(x_t, y_t)\|^2 + \frac{11 L_y^2 \gamma}{4\rho}\mathbb{E}\|y_t - y^*(x_t)\|^2 - \frac{\eta_t \mu \lambda}{4}\mathbb{E}\|y_t - y^*(x_t)\|^2$$

$$- \frac{3\eta_t \lambda^2}{4}\mathbb{E}\|v_t\|^2 + \frac{25\eta_t \lambda}{6\mu}\mathbb{E}\|\nabla_y g(x_t, y_t) - v_t\|^2 + \frac{25\kappa^2 \gamma^2}{6\eta_t \mu \lambda}\mathbb{E}\|\tilde{\mathcal{G}}_t\|^2$$

$$\leq -\Big(\frac{3\gamma\rho}{16} - \frac{25\kappa^2 \gamma^2}{3\eta_t \mu \lambda}\Big)\mathbb{E}\|\mathcal{G}_t\|^2 + \Big(\frac{11\gamma}{4\rho} + \frac{50\kappa^2 \gamma^2}{3\eta_t \mu \lambda \rho^2}\Big)\mathbb{E}\|w_t - \bar{\nabla}f(x_t, y_t)\|^2$$

$$+ \Big(\frac{11 L_y^2 \gamma}{4\rho} + \frac{50\kappa^2 \gamma^2 L_y^2}{3\eta_t \mu \lambda \rho^2} - \frac{\eta_t \mu \lambda}{4}\Big)\mathbb{E}\|y_t - y^*(x_t)\|^2 - \frac{3\eta_t \lambda^2}{4}\mathbb{E}\|v_t\|^2$$

$$+ \frac{25\eta_t \lambda}{6\mu}\mathbb{E}\|\nabla_y g(x_t, y_t) - v_t\|^2, \quad (47)$$

where the last inequality holds by the following inequality

$$\|\tilde{\mathcal{G}}_t\|^2 \leq 2\|\mathcal{G}_t\|^2 + 2\|\tilde{\mathcal{G}}_t - \mathcal{G}_t\|^2$$

$$\leq 2\|\mathcal{G}_t\|^2 + \frac{2}{\rho^2}\|w_t - \nabla F(x_t)\|^2$$

$$\leq 2\|\mathcal{G}_t\|^2 + \frac{4}{\rho^2}\|w_t - \bar{\nabla}f(x_t, y_t)\|^2 + \frac{4 L_y^2}{\rho^2}\|y_t - y^*(x_t)\|^2. \quad (48)$$

Let $\eta = \eta_t$ for all $t \geq 0$. By using $0 < \gamma \leq \frac{9\eta\rho\mu\lambda}{800\kappa^2}$, we have

$$\frac{3\gamma\rho}{32} \geq \frac{50\kappa^2 \gamma^2}{6\eta\lambda\mu}, \quad \frac{3 L_y^2 \gamma}{16\rho} \geq \frac{50\kappa^2 \gamma^2 L_y^2}{3\eta_t \mu \lambda \rho^2}, \quad \frac{3\gamma}{16\rho} \geq \frac{50\kappa^2 \gamma^2}{3\eta_t \mu \lambda \rho^2}. \quad (49)$$

Let $\frac{\eta\mu\lambda}{4} \geq \frac{47 L_y^2 \gamma}{4\rho}$, we have $0 < \gamma \leq \frac{\eta\mu\rho\lambda}{47 L_y^2}$. Given $0 < \gamma \leq \min\big(\frac{9\eta\rho\mu\lambda}{800\kappa^2}, \frac{\eta\mu\rho\lambda}{47 L_y^2}\big)$, we have

$$\Omega_{t+1} - \Omega_t \leq -\frac{3\gamma\rho}{32}\mathbb{E}\|\mathcal{G}_t\|^2 + \frac{47\gamma}{4\rho}\mathbb{E}\|w_t - \bar{\nabla}f(x_t, y_t)\|^2 + \frac{25\eta\lambda}{6\mu}\mathbb{E}\|v_t - \nabla_y g(x_t, y_t)\|^2. \quad (50)$$

Thus, we have

$$\mathbb{E}\|\mathcal{G}_t\|^2$$
$$\leq \frac{32(\Omega_t - \Omega_{t+1})}{3\gamma\rho} + \frac{376}{3\rho^2}\mathbb{E}\|w_t - \bar{\nabla}f(x_t, y_t)\|^2 + \frac{400\eta\lambda}{9\gamma\rho\mu}\mathbb{E}\|v_t - \nabla_y g(x_t, y_t)\|^2$$
$$= \frac{32(\Omega_t - \Omega_{t+1})}{3\gamma\rho} + \frac{376}{3\rho^2}\mathbb{E}\|w_t - \bar{\nabla}f(x_t, y_t) - R(x_t, y_t) + R(x_t, y_t)\|^2 + \frac{400\eta\lambda}{9\gamma\rho\mu}\mathbb{E}\|v_t - \nabla_y g(x_t, y_t)\|^2$$
$$\leq \frac{32(\Omega_t - \Omega_{t+1})}{3\gamma\rho} + \frac{752}{3\rho^2}\mathbb{E}\|w_t - \bar{\nabla}f(x_t, y_t) - R(x_t, y_t)\|^2 + \frac{752}{3\rho^2}\|R(x_t, y_t)\|^2 + \frac{400\eta\lambda}{9\gamma\rho\mu}\mathbb{E}\|v_t - \nabla_y g(x_t, y_t)\|^2$$
$$\leq \frac{32(\Omega_t - \Omega_{t+1})}{3\gamma\rho} + \frac{752\sigma^2}{3\rho^2 b} + \frac{400\eta\lambda\sigma^2}{9\gamma\rho\mu b} + \frac{752}{3\rho^2}\mathbb{E}\|R(x_t, y_t)\|^2, \quad (51)$$

where the last inequality holds by Assumption 6 and $w_t = \bar{\nabla}f(x_t, y_t; \mathcal{B}_t) = \frac{1}{b}\sum_{i \in \mathcal{B}_t} \bar{\nabla}f(x_t, y_t, \bar{\xi}_t^i)$, $v_t = \nabla_y g(x_t, y_t; \mathcal{B}_t) = \frac{1}{b}\sum_{i \in \mathcal{B}_t} \nabla_y g(x_t, y_t, \xi_t^i)$.

Taking average over $t = 0, 2, \cdots, T-1$ on both sides of the above inequality (51), we have

$$
\frac{1}{T} \sum_{t=0}^{T-1} \mathbb{E}\|\mathcal{G}_t\|^2 \leq \frac{32(\Omega_0 - \Omega_T)}{3T\gamma\rho} + \frac{752\sigma^2}{3\rho^2 b} + \frac{400\eta\lambda\sigma^2}{9\gamma\rho\mu b} + \frac{752}{3\rho^2} \frac{1}{T} \sum_{t=0}^{T-1} \mathbb{E}\|R(x_t, y_t)\|^2
$$

$$
= \frac{32(\Phi(x_0) + \|y_0 - y^*(x_0)\|^2)}{3T\gamma\rho} - \frac{32\mathbb{E}(\Phi(x_T) + \|y_T - y^*(x_T)\|^2)}{3T\gamma\rho}
$$

$$
+ \frac{752\sigma^2}{3\rho^2 b} + \frac{400\eta\lambda\sigma^2}{9\gamma\rho\mu b} + \frac{752}{3\rho^2} \frac{1}{T} \sum_{t=0}^{T-1} \mathbb{E}\|R(x_t, y_t)\|^2
$$

$$
\leq \frac{32(\Phi(x_0) - \Phi^*)}{3T\gamma\rho} + \frac{32\Delta}{3T\gamma\rho} + \frac{752\sigma^2}{3\rho^2 b} + \frac{400\eta\lambda\sigma^2}{9\gamma\rho\mu b} + \frac{752}{3\rho^2 T^2}, \tag{52}
$$

where the last inequality holds by Assumption 5 and $\mathbb{E}\|R(x_t, y_t)\| \leq \frac{1}{T}$ for all $t \geq 1$ by choosing $K = \frac{L}{\mu}\log(\frac{LC_{fy}T}{\mu})$.

$\square$

## B.3  Convergence Analysis of the ASBiO-BreD Algorithm

In this subsection, we provide the convergence analysis of our ASBiO-BreD algorithm. When $\mod(t, q) \neq 0$, let $R(x_t, y_t) = \bar{\nabla}f(x_t, y_t) - \bar{\nabla}f(x_t, y_t; \bar{\mathcal{I}}_t)$ for all $t \geq 0$, when $\mod(t, q) = 0$, let $R(x_t, y_t) = \bar{\nabla}f(x_t, y_t) - \bar{\nabla}f(x_t, y_t; \bar{\mathcal{B}}_t)$.

**Lemma 12.** *Suppose the stochastic gradients $v_t$ and $w_t$ be generated from Algorithm 3, we have*

$$
\mathbb{E}\|\bar{\nabla}f(x_t, y_t) + R(x_t, y_t) - w_t\|^2 \leq \frac{2L_K^2}{b_1} \sum_{i=(n_t-1)q}^{t-1} \left(\mathbb{E}\|x_{i+1} - x_i\|^2 + \mathbb{E}\|y_{i+1} - y_i\|^2\right) + \frac{\sigma^2}{b}, \tag{53}
$$

$$
\mathbb{E}\|\nabla_y g(x_t, y_t) - v_t\|^2 \leq \frac{2L^2}{b_1} \sum_{i=(n_t-1)q}^{t-1} \left(\mathbb{E}\|x_{i+1} - x_i\|^2 + \mathbb{E}\|y_{i+1} - y_i\|^2\right) + \frac{\sigma^2}{b}. \tag{54}
$$

*Proof.* We first prove the inequality (53). According to the definition of $w_{t-1}$ in Algorithm 3, we have

$$
w_t - w_{t-1} = \bar{\nabla}f(x_t, y_t; \bar{\mathcal{I}}_t) - \bar{\nabla}f(x_{t-1}, y_{t-1}; \bar{\mathcal{I}}_t). \tag{55}
$$

Then we have

$$
\mathbb{E}\|\bar{\nabla}f(x_t, y_t) + R(x_t, y_t) - w_t\|^2
$$
$$
= \mathbb{E}\|\bar{\nabla}f(x_t, y_t) + R(x_t, y_t) - w_{t-1} - (w_t - w_{t-1})\|^2
$$
$$
= \mathbb{E}\|\bar{\nabla}f(x_t, y_t) + R(x_t, y_t) - w_{t-1} - \bar{\nabla}f(x_t, y_t; \bar{\mathcal{I}}_t) + \bar{\nabla}f(x_{t-1}, y_{t-1}; \bar{\mathcal{I}}_t)\|^2
$$
$$
= \mathbb{E}\|\bar{\nabla}f(x_{t-1}, y_{t-1}) + R(x_{t-1}, y_{t-1}) - w_{t-1} + \bar{\nabla}f(x_t, y_t) + R(x_t, y_t) - \bar{\nabla}f(x_{t-1}, y_{t-1}) - R(x_{t-1}, y_{t-1})
$$
$$
- \bar{\nabla}f(x_t, y_t; \bar{\mathcal{I}}_t) + \bar{\nabla}f(x_{t-1}, y_{t-1}; \bar{\mathcal{I}}_t)\|^2
$$
$$
= \mathbb{E}\|\bar{\nabla}f(x_{t-1}, y_{t-1}) + R(x_{t-1}, y_{t-1}) - w_{t-1}\|^2 + \mathbb{E}\|\bar{\nabla}f(x_t, y_t) + R(x_t, y_t) - \bar{\nabla}f(x_{t-1}, y_{t-1}) - R(x_{t-1}, y_{t-1})
$$
$$
- \bar{\nabla}f(x_t, y_t; \bar{\mathcal{I}}_t) + \bar{\nabla}f(x_{t-1}, y_{t-1}; \bar{\mathcal{I}}_t)\|^2
$$
$$
= \mathbb{E}\|\bar{\nabla}f(x_{t-1}, y_{t-1}) + R(x_{t-1}, y_{t-1}) - w_{t-1}\|^2 + \frac{1}{b_1}\mathbb{E}\|\bar{\nabla}f(x_t, y_t) + R(x_t, y_t) - \bar{\nabla}f(x_{t-1}, y_{t-1}) - R(x_{t-1}, y_{t-1})
$$
$$
- \left(\bar{\nabla}f(x_t, y_t; \xi_t^1) - \bar{\nabla}f(x_{t-1}, y_{t-1}; \xi_t^1)\right)\|^2
$$
$$
\leq \mathbb{E}\|\bar{\nabla}f(x_{t-1}, y_{t-1}) + R(x_{t-1}, y_{t-1}) - w_{t-1}\|^2 + \frac{1}{b_1}\mathbb{E}\|\bar{\nabla}f(x_t, y_t; \xi_t^1) - \bar{\nabla}f(x_{t-1}, y_{t-1}; \xi_t^1)\|^2
$$
$$
\leq \mathbb{E}\|\bar{\nabla}f(x_{t-1}, y_{t-1}) + R(x_{t-1}, y_{t-1}) - w_{t-1}\|^2 + \frac{2L_K^2}{b_1}\left(\|x_t - x_{t-1}\|^2 + \|y_t - y_{t-1}\|^2\right), \tag{56}
$$

where the fourth equality follows by $\mathbb{E}_{\bar{\mathcal{I}}_t}\left[\bar{\nabla}f(x_t, y_t) + R(x_t, y_t) - \bar{\nabla}f(x_{t-1}, y_{t-1}) - R(x_{t-1}, y_{t-1}) - \left(\bar{\nabla}f(x_t, y_t; \bar{\mathcal{I}}_t) - \bar{\nabla}f(x_{t-1}, y_{t-1}; \bar{\mathcal{I}}_t)\right)\right] = 0$; the fifth equality holds by Lemma 8 and $\bar{\nabla}f(x_t, y_t; \bar{\mathcal{I}}_t) = \frac{1}{b_1}\sum_{i\in\bar{\mathcal{I}}_t}\bar{\nabla}f(x_t, y_t; \bar{\xi}_t^i)$, $\bar{\nabla}f(x_{t-1}, y_{t-1}; \bar{\mathcal{I}}_t) = \frac{1}{b_1}\sum_{i\in\bar{\mathcal{I}}_t}\bar{\nabla}f(x_{t-1}, y_{t-1}; \bar{\xi}_t^i)$; the second last inequality holds by the inequality $\mathbb{E}\|\zeta - \mathbb{E}[\zeta]\|^2 \leq \mathbb{E}\|\zeta\|^2$; the last inequality is due to Lemma 4.

Throughout the paper, let $n_t = \lceil t/q \rceil$ such that $(n_t - 1)q \le t \le n_t q - 1$. Telescoping (56) over $t$ from $(n_t - 1)q + 1$ to $t$, we have

$$\mathbb{E}\|\bar{\nabla}f(x_t, y_t) + R(x_t, y_t) - w_t\|^2 \le \frac{2L_K^2}{b_1} \sum_{i=(n_t-1)q}^{t-1} \left(\mathbb{E}\|x_{i+1} - x_i\|^2 + \mathbb{E}\|y_{i+1} - y_i\|^2\right)$$

$$+ \mathbb{E}\|\bar{\nabla}f(x_{(n_t-1)q}, y_{(n_t-1)q}) + R(x_{(n_t-1)q}, y_{(n_t-1)q}) - w_{(n_t-1)q}\|^2$$

$$\le \frac{2L_K^2}{b_1} \sum_{i=(n_t-1)q}^{t-1} \left(\mathbb{E}\|x_{i+1} - x_i\|^2 + \mathbb{E}\|y_{i+1} - y_i\|^2\right) + \frac{\sigma^2}{b}, \qquad (57)$$

where the last inequality is due to Assumption 6 and $w_{(n_t-1)q} = \frac{1}{b}\sum_{i \in \mathcal{B}_{(n_t-1)q}} \bar{\nabla}f(x_{(n_t-1)q}, y_{(n_t-1)q}, \xi^i_{(n_t-1)q})$. Similarly, we can obtain

$$\mathbb{E}\|\nabla_y g(x_t, y_t) - v_t\|^2 \le \frac{2L^2}{b_1} \sum_{i=(n_t-1)q}^{t-1} \left(\mathbb{E}\|x_{i+1} - x_i\|^2 + \mathbb{E}\|y_{i+1} - y_i\|^2\right) + \frac{\sigma^2}{b}. \qquad (58)$$

$\square$

**Theorem 6.** *(Restatement of Theorem 3) Suppose the sequence $\{x_t, y_t\}_{t=1}^T$ be generated from Algorithm 3. Let $b_1 = q$, $K = \frac{L}{\mu}\log(\frac{LC_{fy}T}{\mu})$, $0 < \eta = \eta_t \le 1$, $0 < \gamma \le \min\left(\frac{3\rho}{38L_K^2\eta}, \frac{3\rho}{4L_F}, \frac{2\rho\eta\mu\lambda}{19L_y^2}, \frac{\rho\eta}{8}, \frac{9\rho\eta\mu\lambda}{400\kappa^2}\right)$ and $0 < \lambda \le \min(\frac{1}{6L}, \frac{9\mu}{100\eta^2L^2})$, we have*

$$\frac{1}{T}\sum_{t=0}^{T-1} \mathbb{E}\|\mathcal{G}_t\|^2 \le \frac{32(\Phi(x_0) - \Phi^*)}{3T\gamma\rho} + \frac{32\Delta}{3T\gamma\rho} + \frac{152}{3T^2\rho^2} + \frac{4}{\eta\rho\gamma}\left(\frac{1}{L^2} + \frac{1}{L_K^2}\right)\frac{\sigma^2}{b}, \qquad (59)$$

*where $\Delta = \|y_0 - y^*(x_0)\|^2$.*

*Proof.* This proof is similar to the proof of Theorem 5. According to the above Lemma 2, the function $F(x)$ has $L_F$-Lipschitz continuous gradient. Let $\tilde{\mathcal{G}}_t = \frac{1}{\gamma}(x_t - x_{t+1})$, we have

$$F(x_{t+1}) \le F(x_t) + \langle \nabla F(x_t), x_{t+1} - x_t \rangle + \frac{L_F}{2}\|x_{t+1} - x_t\|^2$$

$$= F(x_t) - \gamma\langle \nabla F(x_t), \tilde{\mathcal{G}}_t \rangle + \frac{\gamma^2 L_F}{2}\|\tilde{\mathcal{G}}_t\|^2$$

$$= F(x_t) - \gamma\langle w_t, \tilde{\mathcal{G}}_t \rangle + \gamma\langle w_t - \nabla F(x_t), \tilde{\mathcal{G}}_t \rangle + \frac{\gamma^2 L_F}{2}\|\tilde{\mathcal{G}}_t\|^2$$

$$\le F(x_t) - \gamma\rho\|\tilde{\mathcal{G}}_t\|^2 - h(x_{t+1}) + h(x_t) + \gamma\langle w_t - \nabla F(x_t), \tilde{\mathcal{G}}_t \rangle + \frac{\gamma^2 L_F}{2}\|\tilde{\mathcal{G}}_t\|^2$$

$$\le F(x_t) + \left(\frac{\gamma^2 L_F}{2} - \frac{3\gamma\rho}{4}\right)\|\tilde{\mathcal{G}}_t\|^2 - h(x_{t+1}) + h(x_t) + \frac{\gamma}{\rho}\|w_t - \nabla F(x_t)\|^2, \qquad (60)$$

where the second last inequality holds by the above Lemma 9, and the last inequality holds by the following inequality

$$\langle w_t - \nabla F(x_t), \tilde{\mathcal{G}}_t \rangle \le \|w_t - \nabla F(x_t)\|\|\tilde{\mathcal{G}}_t\|$$

$$\le \frac{1}{\rho}\|w_t - \nabla F(x_t)\|^2 + \frac{\rho}{4}\|\tilde{\mathcal{G}}_t\|^2. \qquad (61)$$

According to the above Lemma 2, we have

$$\|w_t - \nabla F(x_t)\|^2 = \|w_t - \bar{\nabla}f(x_t, y_t) + \bar{\nabla}f(x_t, y_t) - \nabla F(x_t)\|^2$$

$$\le 2\|w_t - \bar{\nabla}f(x_t, y_t)\|^2 + 2\|\bar{\nabla}f(x_t, y_t) - \nabla F(x_t)\|^2$$

$$\le 2\|w_t - \bar{\nabla}f(x_t, y_t)\|^2 + 2L_y^2\|y_t - y^*(x_t)\|^2. \qquad (62)$$

Let $\Phi(x) = F(x) + h(x)$, plugging (62) into (60), we have

$$\Phi(x_{t+1}) \le \Phi(x_t) + \left(\frac{\gamma^2 L_F}{2} - \frac{3\gamma\rho}{4}\right)\|\tilde{\mathcal{G}}_t\|^2 + \frac{2\gamma}{\rho}\|w_t - \nabla_x f(x_t, y_t)\|^2 + \frac{2L_y^2\gamma}{\rho}\|y_t - y^*(x_t)\|^2$$

$$\le \Phi(x_t) - \frac{3\gamma\rho}{8}\|\tilde{\mathcal{G}}_t\|^2 + \frac{2\gamma}{\rho}\|w_t - \nabla_x f(x_t, y_t)\|^2 + \frac{2L_y^2\gamma}{\rho}\|y_t - y^*(x_t)\|^2$$

$$= \Phi(x_t) - \frac{3\gamma\rho}{16}\|\tilde{\mathcal{G}}_t\|^2 - \frac{3\rho}{16\gamma}\|x_{t+1} - x_t\|^2 + \frac{2\gamma}{\rho}\|w_t - \nabla_x f(x_t, y_t)\|^2 + \frac{2L_y^2\gamma}{\rho}\|y_t - y^*(x_t)\|^2, \qquad (63)$$

where the second last inequality is due to $0 < \gamma \leq \frac{3\rho}{4L_F}$. By using Lemma 10, the difference between $\tilde{\mathcal{G}}_t$ and $\mathcal{G}_t$ are bounded, we have

$$
\begin{aligned}
\|\mathcal{G}_t\|^2 &\leq 2\|\tilde{\mathcal{G}}_t\|^2 + 2\|\tilde{\mathcal{G}}_t - \mathcal{G}_t\|^2 \\
&\leq 2\|\tilde{\mathcal{G}}_t\|^2 + \frac{2}{\rho^2}\|w_t - \nabla F(x_t)\|^2 \\
&\leq 2\|\tilde{\mathcal{G}}_t\|^2 + \frac{4}{\rho^2}\|w_t - \bar{\nabla}f(x_t, y_t)\|^2 + \frac{4L_y^2}{\rho^2}\|y_t - y^*(x_t)\|^2.
\end{aligned}
\tag{64}
$$

Thus we have

$$
-\|\tilde{\mathcal{G}}_t\|^2 \leq -\frac{1}{2}\|\mathcal{G}_t\|^2 + \frac{2}{\rho^2}\|w_t - \bar{\nabla}f(x_t, y_t)\|^2 + \frac{2L_y^2}{\rho^2}\|y_t - y^*(x_t)\|^2.
\tag{65}
$$

By plugging (65) into (63), we have

$$
\begin{aligned}
\Phi(x_{t+1}) &\leq \Phi(x_t) - \frac{3\gamma\rho}{32}\|\mathcal{G}_t\|^2 + \frac{3\gamma\rho}{16}\Big(\frac{2}{\rho^2}\|w_t - \bar{\nabla}f(x_t, y_t)\|^2 + \frac{2L_y^2}{\rho^2}\|y_t - y^*(x_t)\|^2\Big) \\
&\quad - \frac{3\rho}{16\gamma}\|x_{t+1} - x_t\|^2 + \frac{2\gamma}{\rho}\|w_t - \bar{\nabla}f(x_t, y_t)\|^2 + \frac{2L_y^2\gamma}{\rho}\|y_t - y^*(x_t)\|^2 \\
&= \Phi(x_t) - \frac{3\gamma\rho}{32}\|\mathcal{G}_t\|^2 - \frac{3\rho}{16\gamma}\|x_{t+1} - x_t\|^2 + \frac{19\gamma}{8\rho}\|w_t - \bar{\nabla}f(x_t, y_t)\|^2 + \frac{19L_y^2\gamma}{8\rho}\|y_t - y^*(x_t)\|^2.
\end{aligned}
\tag{66}
$$

Next, we define a useful Lyapunov function, for any $t \geq 1$

$$
\Omega_t = \mathbb{E}\big[\Phi(x_t) + \|y_t - y^*(x_t)\|^2\big].
\tag{67}
$$

According to Lemma 11, we have

$$
\begin{aligned}
\|y_{t+1} - y^*(x_{t+1})\|^2 - \|y_t - y^*(x_t)\|^2 &\leq -\frac{\eta_t\mu\lambda}{4}\|y_t - y^*(x_t)\|^2 - \frac{3\eta_t\lambda^2}{4}\|v_t\|^2 \\
&\quad + \frac{25\eta_t\lambda}{6\mu}\|\nabla_y g(x_t, y_t) - v_t\|^2 + \frac{25\kappa^2}{6\eta_t\mu\lambda}\|x_{t+1} - x_t\|^2.
\end{aligned}
\tag{68}
$$

Let $\eta = \eta_t$ for all $t \geq 0$. Then we have

$$
\begin{aligned}
\Omega_{t+1} - \Omega_t &= \mathbb{E}\big[\Phi(x_{t+1}) - \Phi(x_t) + \|y_{t+1} - y^*(x_{t+1})\|^2 - \|y_t - y^*(x_t)\|^2\big] \\
&\leq -\frac{3\gamma\rho}{32}\mathbb{E}\|\mathcal{G}_t\|^2 - \frac{3\rho}{16\gamma}\mathbb{E}\|x_{t+1} - x_t\|^2 + \frac{19\gamma}{8\rho}\mathbb{E}\|w_t - \bar{\nabla}f(x_t, y_t)\|^2 + \frac{19L_y^2\gamma}{8\rho}\mathbb{E}\|y_t - y^*(x_t)\|^2 \\
&\quad - \frac{\eta_t\mu\lambda}{4}\mathbb{E}\|y_t - y^*(x_t)\|^2 - \frac{3\eta_t\lambda^2}{4}\mathbb{E}\|v_t\|^2 + \frac{25\eta_t\lambda}{6\mu}\mathbb{E}\|\nabla_y g(x_t, y_t) - v_t\|^2 + \frac{25\kappa^2}{6\eta_t\mu\lambda}\mathbb{E}\|x_{t+1} - x_t\|^2 \\
&= -\frac{3\gamma\rho}{32}\mathbb{E}\|\mathcal{G}_t\|^2 - \Big(\frac{3\rho}{16\gamma} - \frac{25\kappa^2}{6\eta_t\mu\lambda}\Big)\mathbb{E}\|x_{t+1} - x_t\|^2 + \frac{19\gamma}{8\rho}\mathbb{E}\|w_t - \bar{\nabla}f(x_t, y_t)\|^2 \\
&\quad - \Big(\frac{\eta_t\mu\lambda}{4} - \frac{19L_y^2\gamma}{8\rho}\Big)\mathbb{E}\|y_t - y^*(x_t)\|^2 - \frac{3\eta_t\lambda^2}{4}\mathbb{E}\|v_t\|^2 + \frac{25\eta_t\lambda}{6\mu}\mathbb{E}\|\nabla_y g(x_t, y_t) - v_t\|^2 \\
&\leq -\frac{3\gamma\rho}{32}\mathbb{E}\|\mathcal{G}_t\|^2 - \Big(\frac{3\rho}{16\gamma} - \frac{25\kappa^2}{6\eta_t\mu\lambda}\Big)\mathbb{E}\|x_{t+1} - x_t\|^2 + \frac{19\gamma}{4\rho}\mathbb{E}\|w_t - \bar{\nabla}f(x_t, y_t) - R(x_t, y_t)\|^2 \\
&\quad + \frac{19\gamma}{4\rho}\mathbb{E}\|R(x_t, y_t)\|^2 - \Big(\frac{\eta_t\mu\lambda}{4} - \frac{19L_y^2\gamma}{8\rho}\Big)\mathbb{E}\|y_t - y^*(x_t)\|^2 - \frac{3\eta_t\lambda^2}{4}\mathbb{E}\|v_t\|^2 \\
&\quad + \frac{25\eta_t\lambda}{6\mu}\mathbb{E}\|\nabla_y g(x_t, y_t) - v_t\|^2 \\
&\leq -\frac{3\gamma\rho}{32}\mathbb{E}\|\mathcal{G}_t\|^2 - \Big(\frac{3\rho}{16\gamma} - \frac{25\kappa^2}{6\eta\mu\lambda}\Big)\mathbb{E}\|x_{t+1} - x_t\|^2 + \frac{19\gamma}{4\rho}\mathbb{E}\|w_t - \bar{\nabla}f(x_t, y_t) - R(x_t, y_t)\|^2 \\
&\quad + \frac{19\gamma}{4\rho}\mathbb{E}\|R(x_t, y_t)\|^2 - \frac{3\eta\lambda^2}{4}\mathbb{E}\|v_t\|^2 + \frac{25\eta\lambda}{6\mu}\mathbb{E}\|\nabla_y g(x_t, y_t) - v_t\|^2,
\end{aligned}
\tag{69}
$$

where the last inequality holds by $\gamma \leq \frac{2\rho\eta\mu\lambda}{19L_y^2}$.

Summing over $t = 0, 1, \cdots, T - 1$ on both sides of (69), by Lemma 12, we have

$$\frac{3\gamma\rho}{32} \sum_{t=0}^{T-1} \mathbb{E}\|\mathcal{G}_t\|^2$$

$$\leq \Omega_0 - \Omega_T - \left(\frac{3\rho}{16\gamma} - \frac{25\kappa^2}{6\eta\mu\lambda}\right) \sum_{t=0}^{T-1} \mathbb{E}\|x_{t+1} - x_t\|^2 - \frac{3\eta\lambda^2}{4} \mathbb{E}\|v_t\|^2 + \frac{19\gamma}{4\rho} \sum_{t=0}^{T-1} \mathbb{E}\|R(x_t, y_t)\|^2$$

$$+ \frac{19\gamma}{4\rho} \sum_{t=0}^{T-1} \left(\frac{L_K^2}{b_1} \sum_{i=(n_t-1)q}^{t-1} \left(\mathbb{E}\|x_{i+1} - x_i\|^2 + \mathbb{E}\|y_{i+1} - y_i\|^2\right) + \frac{\sigma^2}{b}\right)$$

$$+ \frac{25\eta\lambda}{6\mu} \sum_{t=0}^{T-1} \left(\frac{L^2}{b_1} \sum_{i=(n_t-1)q}^{t-1} \left(\mathbb{E}\|x_{i+1} - x_i\|^2 + \mathbb{E}\|y_{i+1} - y_i\|^2\right) + \frac{\sigma^2}{b}\right)$$

$$\leq \Omega_0 - \Omega_T - \left(\frac{3\rho}{16\gamma} - \frac{25\kappa^2}{6\eta\mu\lambda}\right) \sum_{t=0}^{T-1} \mathbb{E}\|x_{t+1} - x_t\|^2 - \frac{3\eta\lambda^2}{4} \mathbb{E}\|v_t\|^2 + \frac{19\gamma}{4\rho} \sum_{t=0}^{T-1} \mathbb{E}\|R(x_t, y_t)\|^2$$

$$+ \frac{19\gamma}{4\rho} \sum_{t=0}^{T-1} \left(\frac{L_K^2 q}{b_1}(\mathbb{E}\|x_{t+1} - x_t\|^2 + \mathbb{E}\|y_{t+1} - y_t\|^2) + \frac{\sigma^2}{b}\right)$$

$$+ \frac{25\eta\lambda}{6\mu} \sum_{t=0}^{T-1} \left(\frac{L^2 q}{b_1}(\mathbb{E}\|x_{t+1} - x_t\|^2 + \mathbb{E}\|y_{t+1} - y_t\|^2) + \frac{\sigma^2}{b}\right)$$

$$= \Omega_0 - \Omega_T - \left(\frac{3\rho}{16\gamma} - \frac{25\kappa^2}{6\eta\mu\lambda} - \frac{19\gamma L_K^2 q}{4\rho b_1} - \frac{25\eta\lambda L^2 q}{6\mu b_1}\right) \sum_{t=0}^{T-1} \mathbb{E}\|x_{t+1} - x_t\|^2 + \frac{19\gamma}{4\rho} \sum_{t=0}^{T-1} \mathbb{E}\|R(x_t, y_t)\|^2$$

$$- \left(\frac{3\eta\lambda^2}{4} - \frac{19\gamma L_K^2 q\eta^2\lambda^2}{4\rho b_1} - \frac{25\lambda^3 L^2 q\eta^3}{6\mu b_1}\right) \sum_{t=0}^{T-1} \mathbb{E}\|v_t\|^2 + \left(\frac{19\gamma}{4\rho} + \frac{25\eta\lambda}{6\mu}\right)\frac{T\sigma^2}{b}, \tag{70}$$

where the second inequality holds by $\sum_{t=0}^{T-1} \sum_{i=(n_t-1)q}^{t-1} \left(\mathbb{E}\|x_{i+1} - x_i\|^2 + \mathbb{E}\|y_{i+1} - y_i\|^2\right) \leq q \sum_{t=0}^{T-1} \left(\mathbb{E}\|x_{t+1} - x_t\|^2 + \mathbb{E}\|y_{t+1} - y_t\|^2\right)$.

Let $b_1 = q$, $0 < \gamma \leq \frac{3\rho}{38L_K^2\eta}$ and $0 < \lambda \leq \frac{9\mu}{100\eta^2 L^2}$, we have $\frac{3\eta\lambda^2}{8} \geq \frac{19\gamma L_K^2 q\eta^2\lambda^2}{4\rho b_1}$ and $\frac{3\eta\lambda^2}{8} \geq \frac{25\lambda^3 L^2 q\eta^3}{6\mu b_1}$, i.e., we obtain

$$\frac{3\eta\lambda^2}{4} - \frac{19\gamma L_K^2 q\eta^2\lambda^2}{4\rho b_1} - \frac{25\lambda^3 L^2 q\eta^3}{6\mu b_1} \geq 0. \tag{71}$$

At the same time, we have $\frac{3}{8\eta} \leq \frac{19\gamma L_K^2 q}{4\rho b_1}$, $\frac{3}{8\eta} \leq \frac{25\eta\lambda L^2 q}{6\mu b_1}$, $\frac{3}{8\eta L_K^2} \leq \frac{19\gamma}{4\rho}$ and $\frac{3}{8\eta L^2} \leq \frac{25\eta\lambda}{6\mu}$. Thus we have $\frac{3}{4\eta} \geq \frac{19\gamma L_K^2 q}{4\rho b_1} + \frac{25\eta\lambda L^2 q}{6\mu b_1}$. Let $\gamma \leq \min\left(\frac{\rho\eta}{8}, \frac{9\rho\eta\mu\lambda}{400\kappa^2}\right)$, we have

$$\frac{3\rho}{16\gamma} \geq \frac{25\kappa^2}{6\eta\mu\lambda} + \frac{3}{4\eta} \geq \frac{25\kappa^2}{6\eta\mu\lambda} + \frac{19\gamma L_f^2 q}{8\rho b_1} + \frac{25\eta\lambda L_f^2 q}{6\mu b_1}. \tag{72}$$

Based on the above inequalities (71) and (72), we have

$$\frac{3\gamma\rho}{32} \sum_{t=0}^{T-1} \mathbb{E}\|\mathcal{G}_t\|^2 \leq \Omega_0 - \Omega_T + \frac{19\gamma}{4\rho} \sum_{t=0}^{T-1} \mathbb{E}\|R(x_t, y_t)\|^2 + \frac{3}{8\eta}\left(\frac{1}{L^2} + \frac{1}{L_K^2}\right)\frac{T\sigma^2}{b}, \tag{73}$$

By using the above inequality (73), we have

$$
\begin{aligned}
\frac{1}{T}\sum_{t=0}^{T-1}\mathbb{E}\|\mathcal{G}_t\|^2 &\leq \frac{32(\Omega_0-\Omega_T)}{3\gamma\rho T} + \frac{152}{3T\rho^2}\sum_{t=0}^{T-1}\mathbb{E}\|R(x_t,y_t)\|^2 + \frac{4}{\eta\rho\gamma}\Big(\frac{1}{L^2}+\frac{1}{L_K^2}\Big)\frac{\sigma^2}{b} \\
&= \frac{32(\Phi(x_0)+\|y_0-y^*(x_0)\|^2)}{3T\gamma\rho} - \frac{32\mathbb{E}(\Phi(x_T)+\|y_T-y^*(x_T)\|^2)}{3T\gamma\rho} \\
&\quad + \frac{152}{3T\rho^2}\sum_{t=0}^{T-1}\mathbb{E}\|R(x_t,y_t)\|^2 + \frac{4}{\eta\rho\gamma}\Big(\frac{1}{L^2}+\frac{1}{L_K^2}\Big)\frac{\sigma^2}{b} \\
&\leq \frac{32(\Phi(x_0)-\Phi^*)}{3T\gamma\rho} + \frac{32\Delta}{3T\gamma\rho} + \frac{152}{3T\rho^2}\sum_{t=0}^{T-1}\mathbb{E}\|R(x_t,y_t)\|^2 + \frac{4}{\eta\rho\gamma}\Big(\frac{1}{L^2}+\frac{1}{L_K^2}\Big)\frac{\sigma^2}{b} \\
&\leq \frac{32(\Phi(x_0)-\Phi^*)}{3T\gamma\rho} + \frac{32\Delta}{3T\gamma\rho} + \frac{152}{3T^2\rho^2} + \frac{4}{\eta\rho\gamma}\Big(\frac{1}{L^2}+\frac{1}{L_K^2}\Big)\frac{\sigma^2}{b}, \quad (74)
\end{aligned}
$$

where the last inequality is due to $\mathbb{E}\|R(x_t,y_t)\| \leq \frac{1}{T}$ for all $t \geq 0$ by choosing $K = \frac{L}{\mu}\log(\frac{LC_{fy}T}{\mu})$.

$\square$