# OpenReview forum: "Enhanced Bilevel Optimization via Bregman Distance"
_NeurIPS.cc/2022/Conference — NeurIPS 2022 Accept_

### Official Review · Reviewer_rq9V · 2022-06-22

**Rating:** 6
**Confidence:** 3
**Soundness:** 3 good
**Presentation:** 3 good
**Contribution:** 3 good

**Summary:**

This paper proposes a new class of bilevel optimization (BO) problems both in deterministic and stochastic forms. Compared to the classic BO problem, their outer function has an additional nonsmooth term, which makes their model more general, e.g. it contains the case when we use $L_1$ regularization.

Then three algorithms are proposed. The first two are used to solve deterministic/stochastic BO problems (in the new form) respectively. And the last one is an accelerated version of the second algorithm.

**Questions:**

1) It is not very clear how to compute the partial derivative $\omega_t$ in Algorithm 1 line 8. Maybe more explanation is needed.

2) In Algorithm 2, the usage of $\eta_t$ may need more explanations. Why the step size is $\lambda \eta_t$ however in Algorithm 1 $\lambda$ is used?

3) It is nice that in both the theoretical and experiment parts, the performance of proposed algorithms is better than baseline algorithms. But where does this improvement come from? When compared with baseline algorithms, I guess $h(x) = 0$, then the major difference is the usage of Bregman distance. Does it account for the improvement? If the authors can find more explanations, this work may have more theoretical value.

**Limitations:**

The efficiency of proposed algorithms depends on the choice of $\psi$, which is used to define Bregman distance.  In the experiment part, it is chosen such that the updating of $x$ is very easy, i.e. a closed-form solution exists when $h(x) = 0$.  But when other $\psi$ is chosen or $h(x) \neq 0$, how to efficiently solve the problem to update $x$? The complexity of solving this subproblem seems does not appear in the comparison with other algorithms. Maybe the authors can make it more clear how to solve this subproblem for general $h(x)$.

**Strengths And Weaknesses:**

Strengths: 1) this paper broadens the class of BO problems that appeared in previous literature which leads to the demand for new algorithms (because previous algorithms can not deal with non-smooth outer function); 2) under similar assumptions compared with related works, the algorithms proposed in this paper achieve the best convergence rate with known condition number

Weaknesses: 1) This paper only mentions one circumstance where we need to consider a non-smooth outer function (when we use $L_1$ regularization).  That may narrow the unique field of application of this work, i.e. this work can do but the previous works can not. It is good to mention more examples/applications of using non-smooth objectives.

---

> ### Author Response · Authors · 2022-07-28
> **Responses to Reviewer rq9V' comments**
>
> Thanks so much for your comments.
>
> **Q1**: Weaknesses: 1) This paper only mentions one circumstance…
>
> **R1**: Since the existing gradient-based bilevel algorithms do not focus on solving the bilevel optimization problems with nonsmooth regularization, they only use the sub-gradient descent to solve these nonsmooth bilevel problems. However, our algorithms use more efficient Bregman-distance-based proximal gradient decent to solve them. Please see the additional nonsmooth experimental results: Specifically, we consider the $L_1$ regularized hyper-representation learning problem, i.e. to learn a sparse hyper-representation. We compare our AsBiO-BreD (Algorithm 3) with various baselines. The test accuracy results for 5-way-1-shot (Table 1) and 5-way-5-shot (Table 2) over the Omniglot dataset are summarized in the Table below:
>
> **Table 1**: The 5-way-1-shot case
> | Time  | 20s | 40s | 60s |
> | ------------- | ------------- | ------------- | ------------- |
> | AID_BiO | 0.6509  | 0.7365 | 0.7762 |
> | ITD_BiO | 0.6411 | 0.7210 | 0.7721 |
> | MRBO |  0.6103 |  0.6971 | 0.7519 |
> | FSLA | 0.6539 | 0.7399 | 0.7661 |
> | VRBO | 0.5951| 0.6805| 0.7429 |
> | VR-saBiAdam | 0.6812 | 0.7141 | 0.7523 |
> |AsBiO-BreD| 0.6653 | 0.7403 | 0.7830 |
>
> **Table 2**: The 5-way-5-shot case
> | Time  | 20s | 40s | 60s |
> | ------------- | ------------- | ------------- | ------------- |
> | AID_BiO | 0.8316  | 0.8779 | 0.9032 |
> | ITD_BiO | 0.8131 | 0.8621 | 0.8968 |
> | MRBO |  0.8174 |  0.8634 | 0.8819 |
> | FSLA | 0.7993 | 0.8485 | 0.8824 |
> | VRBO | 0.7730 | 0.8305 | 0.8745 |
> | VR-saBiAdam | 0.7753 | 0.8188 | 0.8640 |
> |AsBiO-BreD| 0.8529 | 0.8967 | 0.9313 |
>
>
> In the final version of our manuscript, we will mention more examples/applications of using non-smooth objectives. For example, Neural Network Architecture Search (NNAS) can be represented as a nonconvex bilevel problem with nonsmooth regularization such as group-Lasso or other structured-Lasso.
>
> **Q2**: It is not very clear how to compute the partial derivative…
>
> **R2**: Thanks for your suggestion. We have detailed the partial derivative $w_t$ in Lemma 5 at the supplementary material. In the final version of our manuscript, we will move this detailed partial derivative $w_t$ to the main body.
>
> **Q3**: In Algorithm 2, the usage of $\eta_t$ may need more explanations…
>
> **R3**: Thanks for your suggestion. In our stochastic algorithms (SBiO-BredD and ASBiO-BredD), we use two learning rates to update the variable $y$. Under this case, we easily provide the error of updated variable $y$ given in Lemma 10 at the supplementary material. In practice, we can more flexibly choose the learning rate for updating variable $y$.
>
> **Q4**: It is nice that in both the theoretical and experiment parts, …then the major difference is the usage of Bregman distance. Does it account for the improvement?...
>
> **R4**: Yes, you are right. In our algorithms, the Bregman distances improve the performances by fitting the geometry of optimization problems based on the proper Bregman functions. In the convergence analysis, the proper strongly-convex Bregman functions used in Bregman distances can reduce the sample complexity of the proposed algorithms.

---

> > ### Comment · Reviewer_rq9V · 2022-08-06
> > **Thank you for your response**
> >
> > I really appreciate the author's response. All my questions are answered.

---

### Official Review · Reviewer_FFsk · 2022-07-10

**Rating:** 7
**Confidence:** 4
**Soundness:** 4 excellent
**Presentation:** 4 excellent
**Contribution:** 3 good

**Summary:**

This paper incorporate the Bregman distance into bilevel optimization and propose three methods (BiO-BreD, SBiO-BredD, ASBiO-BreD) which targets at addressing deterministic and stochastic bilevel problem. Such proposed algorithms have matched best target accuracy $\epsilon$ and improved the condition number $\kappa$ compared with other benchmarks. Meanwhile, such analysis is adaptable for nonsmooth outer function. The experiments also demonstrate the superior performance of proposed algorithms.

**Questions:**

1. It will be better to include explanations for different Lemmas which can help reader to know what is the meaning behind each lemma in main body.

2. It will be better to discuss the technical innovations compared with previous work [21].  Note that [21] refers to different works in main body and supplementary.

3. It will be better to include a short ablation studies to compare current algorithms performances in different settings. (e.g. different batch sizes).

4. The algorithm is analyzed in non-smooth function space while the experiments are based on smooth function. It will be better to try non smooth experiment.

**Limitations:**

Non negative societal impact.

**Strengths And Weaknesses:**

In terms of strengths, the proposed work shows the condition number improvement in terms of convergence analysis. Meanwhile, in different experimental settings, the proposed algorithms have demonstrated its superior performance individually. Both assumptions and convergence analysis are standard and easy to follow.

In terms of weakness, several Lemmas (Lemma2, Lemma4) in main body lack explanations. The theoretical analysis is very standard while it will be better to point out the technical innovations.

---

> ### Author Response · Authors · 2022-07-28
> **Responses to Reviewer FFsk' comments**
>
> Thanks so much for your positive comments.
>
> **R1**: Thanks for your suggestion. In the final version of our manuscript, we will detail the explanations of lemmas. For example, Lemma 2 shows the smoothness of function $F(x)=f(x,y^*(x))$ and Lipschitz continuous of mapping $y^*(x)$. Lemma 4 shows the Lipschitz continuous of the estimated gradient estimator $ \bar{\nabla}f(x,y;\bar{\xi})$ ( or $\bar{\nabla}f(x_t,y_t;\bar{\xi}_t^i$ ) defined in sub-Section 4.2.
>
> **R2**: Thanks for your suggestion. In our algorithms, our key innovation is to use Bregman-distance to update the variable $x$, and apply the strongly-convex Bregman function in the Bregman-distance to improve the sample complexity of our algorithms. In fact, our algorithms are a class of Bragman-distance-based algorithm framework for bilevel optimization, which do not rely on any specific gradient estimators and varaice-reduced techniques. Specifically, in our deterministic algorithm (BiO-BreD), we use the gradient estimator as in [20]. I our stochastic algorithms (SBiO-BredD, ASBiO-BreD), we use the gradient estimator as in [21]. In our ASBiO-BreD algorithm, we use the variance-reduced technique of SPIDER, while [21] uses the momentum-based variance reduced technique of STORM. Clearly, we can also use other gradient estimators and variance-reduced techniques such as STORM to our algorithms.
>
> [20] Bilevel optimization: Convergence analysis and enhanced design, ICML-21;
>
> [21] A near-optimal algorithm for stochastic bilevel optimization via double-momentum, NeurIPS-21.
>
> Moreover, we provide a convergence analysis framework for our algorithms based on a useful and unified potential function $ \Omega_t = \mathbb{E}[F(x_t) + h(x_t) + ||y_t - y^*(x_t)||^2] $.
>
> **R3**: Thanks for your suggestion. In the final version of our manuscript, we will provide a short ablation studies to compare current algorithms performances in different settings. For example, to different batch-sizes, since we use the standard stochastic gradient estimator in our SBiO-BredD algorithm and the variance-reduced technique of SPIDER in our ASBiO-BredD algorithm, they rely on the relatively large batch-size, e.g., in convergence analysis, we choose batch-size $b= 2\kappa^2\epsilon^{-1} $ in our SBiO-BredD algorithm. While the SUSTAIN algorithm in [21] and the MRBO algorithm [37] use the momentum-based variance-reduced technique of STORM, they do not rely on the large batch sizes. Note that in fact, we provide a class of Bregman-distance-based algorithm framework in our paper, so our stochastic algorithms also can use the momentum-based variance-reduced technique of STORM to reduce the large batch size.
>
> **R4**: Thanks for your suggestion. Here, we add solving the nonsmooth bilevel optimization problems. Specifically, we consider the $L_1$ regularized hyper-representation learning problem, i.e. to learn a sparse hyper-representation. We compare our AsBiO-BreD (Algorithm 3) with various baselines. The test accuracy results for 5-way-1-shot (Table 1) and 5-way-5-shot (Table 2) over the Omniglot dataset are summarized in the Table below:
>
> **Table 1**: The 5-way-1-shot case
> | Time  | 20s | 40s | 60s |
> | ------------- | ------------- | ------------- | ------------- |
> | AID_BiO | 0.6509  | 0.7365 | 0.7762 |
> | ITD_BiO | 0.6411 | 0.7210 | 0.7721 |
> | MRBO |  0.6103 |  0.6971 | 0.7519 |
> | FSLA | 0.6539 | 0.7399 | 0.7661 |
> | VRBO | 0.5951| 0.6805| 0.7429 |
> | VR-saBiAdam | 0.6812 | 0.7141 | 0.7523 |
> |AsBiO-BreD| 0.6653 | 0.7403 | 0.7830 |
>
> **Table 2**: The 5-way-5-shot case
> | Time  | 20s | 40s | 60s |
> | ------------- | ------------- | ------------- | ------------- |
> | AID_BiO | 0.8316  | 0.8779 | 0.9032 |
> | ITD_BiO | 0.8131 | 0.8621 | 0.8968 |
> | MRBO |  0.8174 |  0.8634 | 0.8819 |
> | FSLA | 0.7993 | 0.8485 | 0.8824 |
> | VRBO | 0.7730 | 0.8305 | 0.8745 |
> | VR-saBiAdam | 0.7753 | 0.8188 | 0.8640 |
> |AsBiO-BreD| 0.8529 | 0.8967 | 0.9313 |

---

> > ### Comment · Reviewer_FFsk · 2022-08-06
> > **Thanks for your response**
> >
> > Thanks for your response. Questions have been solved.

---

### Official Review · Reviewer_Ra8S · 2022-07-19

**Rating:** 7
**Confidence:** 4
**Soundness:** 3 good
**Presentation:** 4 excellent
**Contribution:** 3 good

**Summary:**

The paper studies the nonconvex outer-objective and strongly convex inner-objective bilevel optimization problem through the lens of Bregmen distance. The paper covers the deterministic optimizaton and stochastic optimization. in both situation, the authors provide the algorithm and its convergence analysis. The theoretical results shows the proposed algorithm with the aid of Bregman distance improve the performance with respect to the condition number $\kappa$ and utilizing the variance reduction technique could further improve the dependency on $\epsilon$ with a order of $\frac{1}{2}$. The numerical experiment further prove the efficiency of the proposed algorithm.

**Questions:**

Q1: On the algorithmic design side, is there any extra effort that we need to make rather than directly replace the exact gradient with the gradient estimations? If it need some specific design, I would like to increase my score.

Q2: On the theoretical side, is there any extra effort other than incorporating the biases introduced by the gradient estimation? I would like to increase my score of rating if so.

**Limitations:**

The numerical verifications of the problem is limited, for each algorithm, we only have one experiment. The experiment shows the proposed algortihms have lower losses. It would be better if the authors provide more experiments. I would like increase my score if the author could provide more experimental results during rebuttal.

**Strengths And Weaknesses:**

The strengths of the paper is obvious that it proposes the algorithms that could improve the theoretical upper-bound of previous state-of-the-art results, the presentation is clear, and the numerical experiment is sound.

The weaknesses mainly because the improvement is predictable. It is well-known that the variance-reduction technique could improve the oder w.r.t. the accuracy $\epsilon$, and the Bregman disctance helps with the condition number $\kappa$.

---

> ### Author Response · Authors · 2022-07-28
> **Responses to Reviewer Ra8S' comments**
>
> Thanks so much for your positive comments.
>
> **Q1**: On the algorithmic design side,...
>
> **R1**: Thanks for your comment. Our algorithms are a class of Bregman-distance-based algorithm framework for bilevel optimization, which do not rely any gradient estimators. Specifically, in our deterministic algorithm (BiO-BreD), we use the gradient estimator as in [20]. I our stochastic algorithms (SBiO-BredD, ASBiO-BreD), we use the gradient estimator as in [21]. Clearly, we can also use other gradient estimators and variance-reduced techniques to our algorithms.
>
> [20] Bilevel optimization: Convergence analysis and enhanced design, ICML-21;
>
> [21] A near-optimal algorithm for stochastic bilevel optimization via double-momentum, NeurIPS-21.
>
> **Q2**: On the theoretical side,...
>
> **R2**: Thanks for your comment. We provide a useful convergence analysis framework for our Bregman-distance-based algorithms. In our convergence analysis, we establish a useful and unified potential function $ \Omega_t = \mathbb{E}[F(x_t) + h(x_t) + ||y_t - y^*(x_t)||^2] $ for our algorithms.
>
> **Q3**: The numerical verification of the problem is limited,...
>
> **R3**: Thanks for your comment. In fact, we have the other experimental results (test accuracy) about Hyper-representation Learning given in supplementary material (at pages 13-14). Here, we also add solving the nonsmooth bilevel optimization problems. Specifically, we consider the $L_1$ regularized hyper-representation learning problem, i.e. to learn a sparse hyper-representation. We compare our AsBiO-BreD (Algorithm 3) with various baselines. The test accuracy results for 5-way-1-shot (Table 1) and 5-way-5-shot (Table 2) over the Omniglot dataset are summarized in the Table below:
>
> **Table 1**: The 5-way-1-shot case
> | Time  | 20s | 40s | 60s |
> | ------------- | ------------- | ------------- | ------------- |
> | AID_BiO | 0.6509  | 0.7365 | 0.7762 |
> | ITD_BiO | 0.6411 | 0.7210 | 0.7721 |
> | MRBO |  0.6103 |  0.6971 | 0.7519 |
> | FSLA | 0.6539 | 0.7399 | 0.7661 |
> | VRBO | 0.5951| 0.6805| 0.7429 |
> | VR-saBiAdam | 0.6812 | 0.7141 | 0.7523 |
> |AsBiO-BreD| 0.6653 | 0.7403 | 0.7830 |
>
> **Table 2**: The 5-way-5-shot case
> | Time  | 20s | 40s | 60s |
> | ------------- | ------------- | ------------- | ------------- |
> | AID_BiO | 0.8316  | 0.8779 | 0.9032 |
> | ITD_BiO | 0.8131 | 0.8621 | 0.8968 |
> | MRBO |  0.8174 |  0.8634 | 0.8819 |
> | FSLA | 0.7993 | 0.8485 | 0.8824 |
> | VRBO | 0.7730 | 0.8305 | 0.8745 |
> | VR-saBiAdam | 0.7753 | 0.8188 | 0.8640 |
> |AsBiO-BreD| 0.8529 | 0.8967 | 0.9313 |

---

> > ### Comment · Reviewer_Ra8S · 2022-08-07
> > **Thank you for the response**
> >
> > Thank you for the response. You resolve my questions, but I want to remain my score.

---

### Meta-Review · Area_Chair_HbCB · 2022-08-26

**Recommendation:** Accept
**Confidence:** Certain

**Metareview:**

The paper studies bilevel optimization problems, provides three algorithms for different settings, and improves the convergence analysis in terms of the condition number. In addition, numerical experiments are conducted that provide illustration of the effectiveness of the algorithms. Three reviewers all agree that the paper should be published as it contributes to the literature and will be of interest to the NeurIPS audience.

When preparing the final version of the manuscript, please incorporate the discussion that addressed the reviewers' comments either in the main text or the appendix.

**Award:**

No

---

### Decision · Program_Chairs · 2022-09-14

Accept